# A Graph Enhanced Symbolic Discovery Framework for Efficient Logic Optimization

**Yinqi Bai[1]\*, Jie Wang[1]†, Lei Chen[2], Zhihai Wang[1], Yufei Kuang[1], Mingxuan Yuan[2], JianYe Hao[2,3], Feng Wu[1]**

[1]MoE Key Laboratory of Brain-inspired Intelligent Perception and Cognition, University of Science and Technology of China
[2] Noah's Ark Lab, Huawei Technologies
[3] College of Intelligence and Computing, Tianjin University

## Abstract

The efficiency of Logic Optimization (LO) has become one of the key bottlenecks in chip design. To prompt efficient LO, previous studies propose using a key scoring function to predict and prune a large number of ineffective nodes of the LO heuristics. However, the existing scoring functions struggle to balance inference efficiency, interpretability, and generalization performance, which severely hinders their application to modern LO tools. To address this challenge, we propose a novel data-driven circuit symbolic learning framework, namely CMO, to learn lightweight, interpretable, and generalizable scoring functions. The major challenge of developing CMO is to discover symbolic functions that can well generalize to unseen circuits, i.e., the circuit symbolic generalization problem. Thus, the major technical contribution of CMO is the novel *Graph Enhanced Symbolic Discovery* framework, which distills dark knowledge from a well-designed graph neural network (GNN) to enhance the generalization capability of the learned symbolic functions. To the best of our knowledge, CMO is *the first* graph-enhanced approach for discovering lightweight and interpretable symbolic functions that can well generalize to unseen circuits in LO. Experiments on three challenging circuit benchmarks show that the *interpretable* symbolic functions learned by CMO outperform previous state-of-the-art (SOTA) GPU-based and human-designed approaches in terms of *inference efficiency* and *generalization capability*. Moreover, we integrate CMO with the Mfs2 heuristic—one of the most time-consuming LO heuristics. The empirical results demonstrate that CMO significantly improves its efficiency while keeping comparable optimization performance when executed on a CPU-based machine, achieving up to $2.5\times$ faster runtime.

## 1 Introduction

The modern chip design workflow has incorporated multiple Electronic Design Automation (EDA) tools to synthesize, simulate, test, and verify different circuit designs efficiently and reliably (Huang et al., 2021). Logic Optimization (LO) is one of the most important EDA tools in the front-end workflow (Berndt et al., 2022; Pasandi et al., 2023). A key task in LO is Circuit Optimization, which aims to optimize circuits—modeled by directed acyclic graphs—with functionality-equivalent transformations and reduced size and/or depth. It is crucial to well tackle the LO task as it can significantly improve the Quality of Results (QoR), i.e., various metrics that evaluate the quality of designed chips, such as size, level, and edge (De Abreu et al., 2021; Bertacco et al., 1997). However, the LO task can be extremely hard to tackle as it is a $\mathcal{NP}$-hard problem (Micheli, 1994; Farrahi & Sarrafzadeh, 1994). To approximately tackle the LO task, many effective LO heuristics such as Mfs2 (Mishchenko et al., 2011), Resub (Brayton, 2006), and Rewrite (Bertacco et al., 1997) have been developed. Specifically, these heuristics usually apply transformations to subgraphs rooted at each node—that is, the node-level transformations—sequentially for all nodes on the input circuit.

---

\*This work was done when Yinqi Bai was an intern at Huawei Noah's Ark Lab.
†Corresponding author. Email: jiewangx@ustc.edu.cn.

The efficiency of LO heuristics in LO tools has become one of the key bottlenecks in chip design, thus significantly impacting the final circuit performance, power, area (PPA), and Time-to-Market, i.e., the overall duration for developing and commercializing new chips (Neto et al., 2021; Sabbavarapu et al., 2014; Reddy et al., 2014). However, executing LO heuristics can be highly time-consuming due to a large number of ineffective and redundant transformations. To prompt efficient LO, previous works propose a prediction and prune framework, which applies a key scoring function to reduce a large number of ineffective node-level transformations (Li et al., 2023; Wang et al.).

However, the existing scoring functions struggle to balance inference efficiency, interpretability, and generalization performance, which severely hinders their application to modern LO tools. First, (Wang et al.) proposes using a well-designed graph neural network (GNN) model (Kipf & Welling, 2017; Shi et al., 2023) as the scoring function, which offers a promising approach to well tackle the LO task. However, the limited inference efficiency of deep learning models leads to higher runtime costs for LO heuristics, as most current industrial LO tools are purely CPU-based. Moreover, the 'black-box' nature of deep learning methods raises concerns among researchers about the reliability of deploying such models in practical applications. In contrast, (Li et al., 2023) proposes a human-designed hard-coded mathematical expression as the scoring function, which aligns with human intuition and is thus regarded to be reliable. However, designing and developing these functions is extremely challenging as it requires extensive expert knowledge. Moreover, this function cannot achieve high generalization performance due to the lack of adoption of machine learning from existing data , which could significantly degrade the QoR of the optimized circuits.

To address the aforementioned challenges, we propose a novel data-driven circuit symbolic learning framework, namely CMO, which learns a symbolic scoring function that combines the advantages of the two paradigms. An appealing feature of the CMO is its greater potential to discover lightweight and interpretable symbolic functions from a decomposed symbolic space. However, we found that the learned symbolic functions cannot generalize well to unseen circuits, i.e., the circuit symbolic generalization problem. The poor generalization performance could significantly degrade the optimization performance compared to default heuristics. Thus, the major challenge of developing CMO is how to learn symbolic functions that can well generalize to unseen circuits.

To enhance generalization, the major technical contribution of CMO is the novel **G**raph **E**nhanced **S**ymbolic **D**iscovery (GESD) framework, which leverages a well-designed GNN to guide the generation of symbolic trees. Specifically, GESD establishes a teacher-student framework in which the GNN functions as the teacher while a Monte-Carlo Tree Search (MCTS) based symbolic learning model serves as the student. The core idea of GESD is to utilize the teacher GNN's output, which encapsulates domain-invariant information crucial for circuit generalization (Wang et al.), to strategically guide the generation of the student symbolic trees. Consequently, GESD is adept at discovering symbolic functions with strong generalization capabilities. Experiments on the open-source and industrial benchmarks show that the symbolic functions learned by CMO outperform previous state-of-the-art (SOTA) GPU-based and human-designed approaches in terms of inference efficiency and generalization capability. Moreover, we incorporate CMO with the Mfs2 heuristic—the most time-consuming one among commonly used LO heuristics. The empirical results on very large-scale circuits demonstrate that CMO achieves up to 2.5× faster runtime compared with the default Mfs2 heuristic while keeping comparable optimization performance. Furthermore, our GESD learned symbolic functions are all concise expressions that exhibit good interpretability.

We summarize our major contributions as follows: (1) we propose a novel circuit symbolic learning framework, namely CMO, to learn efficient, interpretable, and generalizable symbolic functions that are reliable and simple to deploy in modern LO tools. (2) The major technical contribution of CMO is the novel graph-enhanced symbolic discovery framework which employs a well-designed GNN to enhance the generalization capability of the learned symbolic functions. (3) To the best of our knowledge, CMO is *the first* graph-enhanced approach for discovering lightweight and interpretable symbolic functions that can well generalize to unseen circuits in LO. (4) Experiments demonstrate that the *interpretable* learned symbolic functions outperform previous SOTA GPU-based and human-designed approaches in terms of *inference efficiency* and *generalization capability*. Moreover, it significantly improves the efficiency of the Mfs2 heuristic with comparable optimization performance on a purely CPU-based machine, achieving up to 2.5× faster runtime.

## 2 BACKGROUND

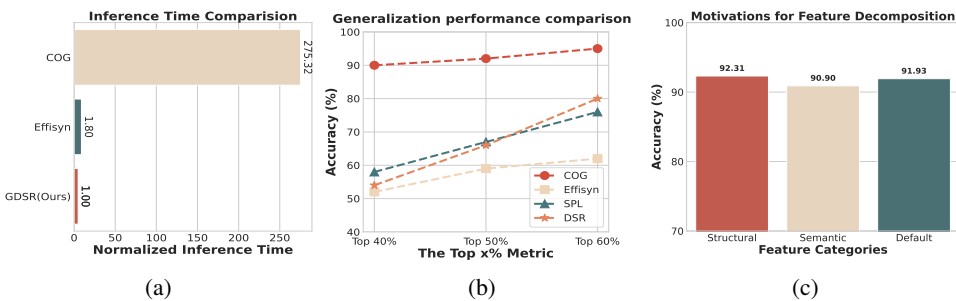

Figure 1: (a) The inference time of the GNN model is significantly larger than other approaches when executed on CPU-based machines. (b) The human-designed approach and existing SOTA symbolic learning approaches cannot generalize well to unseen circuits, i.e., the circuit symbolic generalization problem, while the GNN model exhibits good generalization capability. (c) The circuit node features exhibit separability without compromising the predictive performance.

**The prediction and prune framework for LO Heuristics** Many effective LO heuristics have been developed to tackle the LO task. These LO heuristics follow the same paradigm as shown in Figure 5. Specifically, they apply transformations to subgraphs rooted at each node (i.e., the node-level transformations) sequentially for all nodes on an input circuit. Note that the major differences among these heuristics lie in the node-level transformation mechanism. Recently, (Wang et al.) found that a large number of node-level transformations in many LO heuristics are ineffective, which makes applying these heuristics highly time-consuming. To address this challenge, (Wang et al.) proposes a prediction and prune framework that incorporates a scoring function to select those nodes with top $k$ scores as predicted effective samples and avoid applying transformations on those ineffective samples to improve the efficiency of LO heuristics. However, the scoring function could significantly degrade the optimization performance compared to default heuristics when inaccurately classifying effective nodes in unseen circuits. Therefore, it is essential for the scoring function to achieve high generalization performance for comparable optimization performance.

**Monte Carlo Tree Search (MCTS)** MCTS is an effective method for sequential decision-making problems (Świechowski et al., 2023; Geng et al., 2024). It builds a search tree of possible states and uses stochastic simulations to assess node values, allowing efficient exploration of complex decision spaces. The MCTS algorithm follows four steps: **(a) Selection**: The agent navigates through the search tree based on a policy until it reaches an expandable or terminal node. **(b) Expansion**: At an expandable node, a new child node is added to the tree. **(c) Simulation**: The agent runs simulations from the current node to a terminal state. **(d) Backpropagation**: The results are used to update the statistics of nodes along the path to the root. To balance exploration and exploitation, MCTS always selects the action based on the Upper Confidence Bound for Trees (UCT) (Kocsis & Szepesvári, 2006): $UCT(s, a) = Q(s, a) + c\frac{\ln[N(s)]}{N(s,a)}$. where $Q(s, a)$ is the average reward for action $a$ in state $s$, $N(s)$ is the visit count of $s$, and $N(s, a)$ is the count for choosing $a$ in $s$.

## 3 MOTIVATING RESULTS

**Problem Challenge: Limitations For Existing Scoring Functions** To learn a scoring function that can accurately identify ineffective transformations for efficient LO, (Wang et al.) proposes a well-designed graph neural network, i.e., COG. However, we observe two limitations that severely limit their wide application to modern LO tools:

- Limited inference efficiency on CPUs. We compare the inference time of COG, the human-designed approach Effisyn (Li et al., 2023), and our symbolic discovery method GESD on a CPU-based machine. The results in Figure1a support the observation above.

- The 'black nature' of the learned scoring function. Deep learning models like GNN are often considered uninterpretable, which hinders a deeper understanding of the learned functions and raises safety concerns among researchers.

To alleviate the two aforementioned limitations, a one-step further idea is to employ lighter models. (Li et al., 2023) proposes a human-designed nonlinear scoring function to replace the time-consuming GNNs. However, this method faces a significant limitation:

- The poor generalization performance. The results in Figure1b demonstrate that the human-designed approach struggles to achieve high generalization performance on the open-source and industrial benchmarks, which significantly degrades the QoR of the optimized circuits.

Therefore, how to find a lightweight, interpretable, and generalizable scoring function is the key problem for efficient LO.

**Technical Challenge: The Circuit Symbolic Generalization Problem in LO** In this subsection, we first illustrate the motivation for symbolic discovery. Then we provide a detailed description of the circuit symbolic generalization problem in LO.

Firstly, we found that lightweight and high-performance symbolic functions exist in the circuit data. Specifically, we employ a state-of-the-art (SOTA) symbolic learning approach (Sun et al., 2023) on a widely-used benchmark. The results in Figure 6 demonstrate that among all of the 15 circuits, the prediction recall of 14 circuits exceeds 99%, which further demonstrates the existence of high-performance symbolic functions. Moreover, the results in Figure 1a show that the learned symbolic functions are very efficient. Therefore, there exist lightweight and high-performance symbolic functions in our LO task, which motivate our circuit symbolic learning framework.

However, we found it challenging to learn a symbolic function that can well generalize to unseen circuits from the training dataset. Specifically, we evaluate a well-designed GNN (Wang et al.) and two SOTA symbolic learning methods, i.e., SPL (Sun et al., 2023) and DSR (Petersen et al., 2020) on the open-source and industrial benchmarks under the same generalization evaluation strategy (see Section 5). The results in Figure 1b show that the predictive performance of the learned symbolic functions is significantly lower than that of the GNN, demonstrating the poor generalization capabilities of existing symbolic learning approaches. Thus, the major challenge of developing CMO is to learn symbolic functions that can well tackle the circuit symbolic generalization problem.

## 4 METHOD

In this section, we first provide a detailed description of the novel data-driven circuit symbolic learning framework (CMO) (see Figure 2). Then, we present our graph enhanced symbolic discovery (GESD) framework that can enhance the generalization ability of the learned symbolic functions via graph distillation (see Figure 3). Finally, we demonstrate how we deploy the learned lightweight and interpretable symbolic functions in modern LO tools.

### 4.1 THE DATA-DRIVEN CIRCUIT SYMBOLIC LEARNING FRAMEWORK

As shown in Figure 1a and Figure 1b, we found that existing GNN-based and human-designed approaches struggle to balance inference efficiency, interpretability, and generalization performance. To address this problem, we propose a novel data-driven circuit symbolic learning framework, namely CMO, to discover symbolic functions that combine the advantages of the two paradigms.

**Data Collection** To generate the circuit dataset, we apply a LO heuristic to optimize the circuit, which applies various node-level transformations to each node in the input circuit. For each node in the circuit, we generate a data pair $(\mathbf{x}_i, y_i)$, where $\mathbf{x}_i \in \mathbb{R}^d$ denotes the $i$-th node feature and $y_i \in \mathbb{R}$ denotes the corresponding label. Specifically, if the node-level transformation is effective at the node $\mathbf{x}_i$, then we set $y_i = 1$. Otherwise, $y_i = 0$. Given the LO heuristic and $N$ samples, we generate a dataset $\mathcal{D} = \{\mathbf{x}_i, y_i\}_{i=1}^N$ and aim to find a lightweight and interpretable symbolic function from this dataset that can well generalize to unseen circuits.

**Structural-Semantic Feature Decomposition** Given an input circuit, we choose the node features that contain as much as possible information about the LO task. Therefore, we use the feature vector employed in (Wang et al.), which contains abundant structural and semantic (i.e., functionality) information of the circuit nodes (see Appendix E.4). However, the high dimensionality of the applied features increases the search space and makes it challenging to discover lightweight symbolic functions that effectively capture information from the circuit data. To address this challenge, we propose a structural-semantic feature decomposition mechanism, which separates the node features

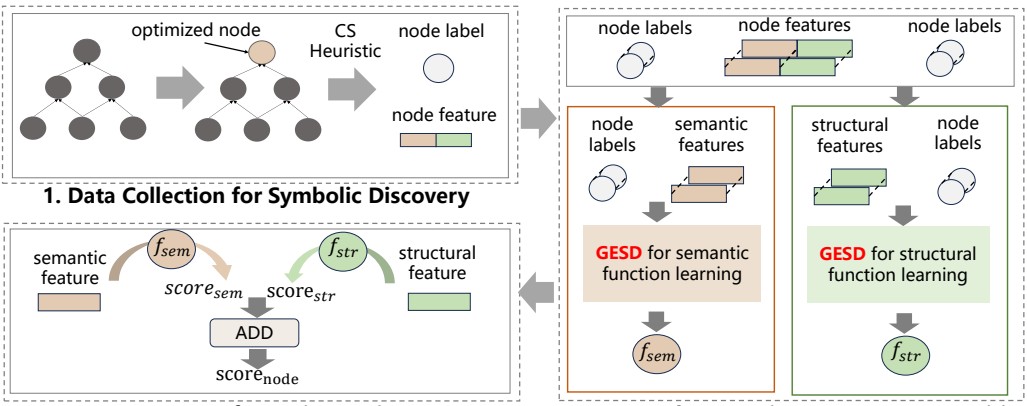

Figure 2: Illustration of the novel data-driven circuit symbolic learning Framework.

$\mathbf{x}_i$ into structural ($\mathbf{x}_i^{str}$) and semantic ($\mathbf{x}_i^{sem}$) components for symbolic discovery. The rationale behind this feature decomposition is based on a key observation—the circuit features exhibit separability without compromising the predictive performance of the scoring model as shown in Figure 1c. Specifically, we decompose the node features into structural and semantic components to train the GNN model, and the results show that the GNN trained on decomposed features could achieve comparable prediction performance to that trained on default features. By applying this decomposition, the original sixty-nine feature variables are reduced to five structural features, thereby significantly shrinking the symbolic search space and enabling lightweight symbolic function discovery.

**GESD for Symbolic Function Learning** After decomposing the init feature into structural and semantic components, we collect structural data $\mathcal{D}_{str} = \{\mathbf{x}_i^{str}, y_i\}_{i=1}^N$ and semantic data $\mathcal{D}_{sem} = \{\mathbf{x}_i^{sem}, y_i\}_{i=1}^N$, where $\mathbf{x}_i^{str}$ refers to structural node feature and $\mathbf{x}_i^{sem}$ refers to semantic node feature. To capture structural information, we employ our Graph Enhanced Symbolic Discovery (GESD) framework to learn a mathematical symbolic function $f^{str} : \mathbf{R}^d \to \mathbf{R}$ (see Section 4.2), as the values of structural features can be approximated as continuous data, making them suitable for continuous mathematical symbolic regression. In contrast, learning mathematical functions for semantic information is challenging due to the discrete and binary nature of both feature values and labels. Thus, to capture semantic information, we formulate the semantic function as a Boolean symbolic learning problem, i.e., employing our GESD framework to learn a boolean function $f^{sem} : \mathbf{B}^d \to \mathbf{B}$ that can accurately identify the effective nodes, where $\mathbf{B} = \{0, 1\}$ denotes the boolean domain.

**Feature Information Fusion** After discovering the structural and functional symbolic functions, we get the score $s_i$ for a node in an unseen circuit using the following equation:

$$\mathbf{s}_i = f^{str}(\mathbf{x}_i^{str}) + w * f^{sem}(\mathbf{x}_i^{sem}) \tag{1}$$

$w$ is a weight value in the range [0, 1] that balances the two types of information. Due to limited space, please refer to Algorithm 3 for a detailed explanation of the derivation process of $\mathbf{s}_i$.

### 4.2 GRAPH ENHANCED SYMBOLIC DISCOVERY FRAMEWORK

To address the circuit symbolic generalization problem (shown in Figure 1b), we introduce the Graph Enhanced Symbolic Discovery (GESD) framework (see Figure 3) which incorporates the strong generalization capability of a well-designed GNN model into our symbolic tree searching process. Note that both the structural and semantic functions follow the same symbolic discovery framework, differing only in training details. Therefore, we provide the implementation details of the semantic symbolic learning process in Appendix E.3.

**Symbolic Operators** In our task, we use an expression tree to express the symbolic functions. The mathematical operators used to generate the expression tree are $\{+, -, \times, \div, \log, \exp, \sin, \cos\}$. We didn't employ placeholders to generate constants while introducing an inner optimization loop generally requires higher training costs (Sun et al., 2023; Xu et al., 2024). Moreover, we found that some complex operators like $\exp, \sin, \cos$ could achieve high prediction performance for the five dimension structural features. Therefore, we use the aforementioned operators.

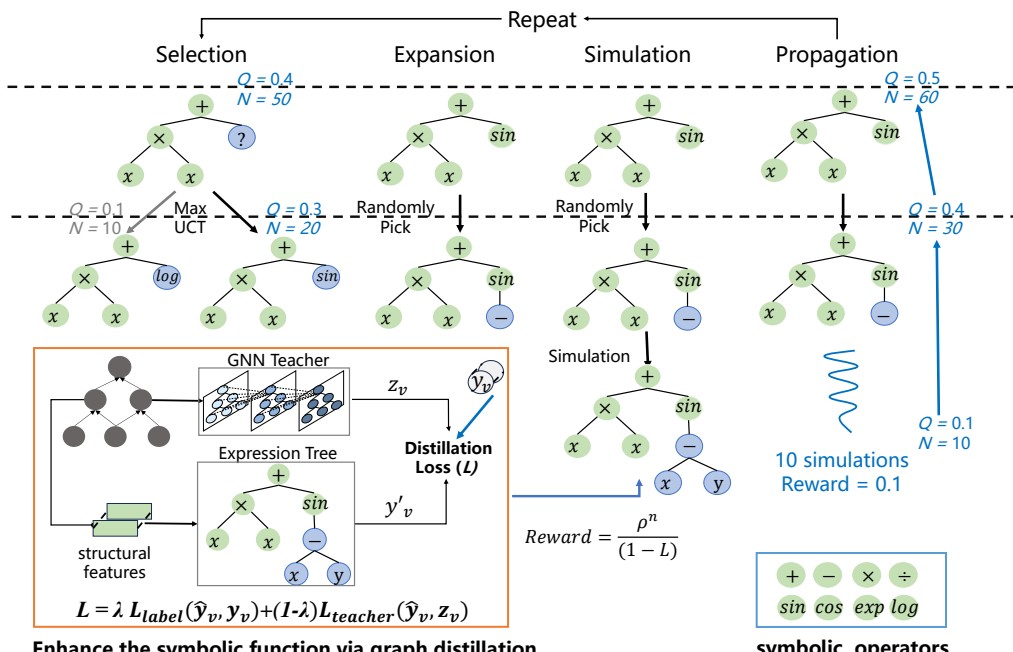

Figure 3: Overview of GESD. GESD employs a Monte Carlo Tree Search algorithm to search in the large and discrete symbolic space. The major novelty of GESD is that we distill dark knowledge from a well-designed GNN into the symbolic tree searching process, thus enhancing the generalization capability of the learned symbolic functions.

**Generator of the Symbolic Tree** Motivated by the strengths of Monte Carlo Tree Search (MCTS) in efficiently exploring large and complex symbolic spaces (Sun et al., 2023; Xu et al., 2024), we employ MCTS to generate symbolic trees. We define the state $s$ as the pre-order traversal of the current expression tree and action $a$ as the symbolic operators or variables added to the state. Specifically, our MCTS includes four steps: selection, expansion, simulation, and propagation. **(1) Selection.** In the selection phase, our MCTS agent traverses the current expression tree and selects an action with the maximum UCT. To ensure the legalization of generated expressions, at the current state $s_t = [a_1, a_2, \ldots, a_t]$, the MCTS agent masks out the invalid action for the current non-terminal node and on that basis selects a valid action as action $a_{t+1}$. **(2) Expansion.** Once the selection phase reaches an expandable node—a node that not all of its children have been visited—our MCTS agent expands it by randomly selecting one of its unvisited valid children. **(3) Simulation**. Given the current state and the expanded node, we perform simulations by randomly selecting the next node until the expression tree is completed. Specifically, we perform 10 simulations and return the maximum simulated reward rather than the average reward for $Q(s, a)$ to find the unique optimal symbolic solution, which is a greedy search heuristic different from traditional MCTS algorithm (Świechowski et al., 2023). **(4) Backpropagation** After the simulation, we update the maximum rewards $Q$ and visited time of nodes $N$ along the path from the current node to the root. The search algorithm repeatedly cycles through the aforementioned steps until the stopping criterion is met.

**Reward Function** To evaluate the symbolic function projected from an expression tree in the simulation phase, we define a reward $r$ based on the function and circuit data. Specifically, we define the reward as $r = \frac{\eta^n}{(1-L)}$. Here, $L$ is the loss function shown in eq 2. To maintain the conciseness of the learned symbolic functions for high efficiency and good interpretability, we follow (Sun et al., 2023) to incorporate a penalty factor into the reward. Specifically, $\eta$ denotes the penalty constant range in $(0, 1)$ and $n$ is the length of the pre-order traversal of the current expression tree.

**Enhancing the symbolic function via graph distillation** Previous studies (Wang et al.) proposed a carefully designed, complex GNN model that effectively addresses the challenge of poor generalization caused by large distribution shifts in circuit domains. However, we found it hard for a symbolic function to capture this crucial generalization information, i.e., the circuit symbolic generalization problem (shown in Figure 1b). The main reason is that, to maintain the simplicity of the learned symbolic functions, we use node features as input instead of subgraphs, which inherently contain

less domain-invariant information. To address this challenge, we establish a teacher-student framework that distills the dark knowledge, i.e., the generalization information, from the teacher GNN's output to strategically guide the student symbolic tree generation. Specifically, we first train a GNN on the training dataset and then employ the prediction output ($\hat{y}$), the output of GNN ($z$), and true labels ($y$) to evaluate every complete expression tree. The evaluation metric is defined as

$$L = \lambda L_{\text{label}}(\hat{y}, y) + (1 - \lambda) L_{\text{teacher}}(\hat{y}, z) \tag{2}$$

$L_{label}$ is introduced to incorporate label information, while $L_{teacher}$ is used to bring in teacher knowledge, providing additional guidance for symbolic function learning. Different from previous works which use KL-divergence for $L_{\text{teacher}}$ (Zhang et al., 2022) to learn the distribution of the teacher's output, we employ the mean squared error (MSE) to directly learn the generalization information from the GNN. The key motivation lies in the discovery of a simple nonlinear mapping between circuit features and the GNN's output, which achieves a generalization performance comparable to that of the GNN itself (see Figure 8). Moreover, note that the number of effective nodes is far fewer than ineffective nodes, which brings a severe imbalance between positive and negative samples in the circuit data. Therefore, we follow (Wang et al.) to leverage the focal loss (Lin et al., 2017) as the $L_{student}$. Due to limited space, please refer to Appendix E.5 for more implementation details of our graph distillation approach.

## 4.3    THE DEPLOYMENT TO MODERN LO TOOLS

After the training process, we obtain a pair of symbolic functions that score and prune the effective transformations. Intuitively, this symbolic function can be regarded as a data-driven version of the human-designed function (Effisyn) in modern LO tools. Moreover, our learned symbolic functions are concise one-line functions that are easy and reliable for deployment. Thus, similar to Effisyn (Li et al., 2023), we directly compile the learned policy to a lightweight shared object using a simple script and then integrate it into the LO tools package.

## 5    EXPERIMENT

We conduct extensive experiments to evaluate CMO, which have four main parts: **Experiment 1.** To demonstrate the superior performance of our CMO in terms of generalization performance and efficiency. **Experiment 2.** To demonstrate that our approach can not only prompt the efficiency of the Mfs2 heuristic but also improve the Quality of Results (QoR). **Experiment 3.** Perform carefully designed ablation studies to provide further insight into CMO. **Experiment 4.** To show the appealing features of CMO in terms of online inference efficiency and interpretability.

**Benchmarks** We evaluate CMO on two widely-used public benchmarks, EPFL (Amarú et al., 2015) and IWLS (Albrecht, 2005), and one industrial benchmark from an anonymous semiconductor company. These benchmarks consist of 69 circuits in total, including very large-scale circuits with up to twenty million nodes. Due to limited space, we defer more details to Appendix D.1.

**Experimental setup** Throughout all experiments, we use ABC (Brayton et al., 2010) as the backend LO framework. ABC is a state-of-the-art open-source LO framework and is widely used in research of machine learning for LO (Pasandi et al., 2023). Moreover, we choose the Mfs2 (Mishchenko et al., 2011)—one of the most time-consuming LO heuristics—as the backend heuristic to optimize. The teacher graph learning model for symbolic distillation is a well-designed 2-layer graph convolutional neural network (GCNN) (Wang et al.). Experiments are performed on a single machine that contains 32 Intel XeonR E5-2667 v4 CPUs. More details are provided in Appendix E.1.

**Evaluation Metrics and Evaluated Methods** Throughout all experiments, we evaluate our method in two separate phases, i.e., the offline and online phases. **In the offline phase**, we evaluate the prediction recall of the effective nodes. We empirically show that the QoR improves with increased prediction recall in Appendix C.2. Thus, it is important to achieve high recall for comparable QoR with the default heuristics. Specifically, we present details as follows. (1) **Evaluation metrics** Under the prediction and prune framework (see Figure 5), we view the prediction task as a scoring task and predict nodes with top $k$ scores to be positive. Under this prediction, we define a **top k accuracy metric** by the fraction of true positive nodes that are predicted to be positive, i.e., prediction recall. We defer details on this metric to Appendix E.1.2. (2) **Evaluated methods** In the offline phase, we evaluate the well-designed GNN approach COG (Wang et al.), the human-designed approach Effisyn (Li et al., 2023), and our CMO. COG is a well-designed 2-layer graph convolutional neural network. Effisyn is a human-designed nonlinear function with key parameters manually derived from circuit

Table 1: Our CMO achieves comparable prediction recall with the well-designed GNN, i.e. COG, and significantly outperforms the human-designed approach, i.e., Effisyn.

| Open-source Circuits | Hyp | Multiplier | Square | DesPerf | Ethernet | Conmax |
|---|---|---|---|---|---|---|
| Method | Recall ↑ | Recall ↑ | Recall ↑ | Recall ↑ | Recall ↑ | Recall ↑ |
| COG | 0.99 | **0.97** | **1.00** | 0.74 | 0.68 | **0.92** |
| Effisyn | 0.94 | 0.44 | 0.46 | 0.51 | 0.32 | 0.68 |
| CMO (Ours) | **0.99** | **0.97** | 0.98 | **0.80** | **0.72** | 0.85 |
| **Very Large-scale & Industrial Circuits** | Sixteen | Twenty | Ci1 | Ci2 | Ci3 | Ci4 |
| Method | Recall ↑ | Recall ↑ | Recall ↑ | Recall ↑ | Recall ↑ | Recall ↑ |
| COG | **0.86** | **0.90** | 0.89 | 0.85 | **1.00** | 0.94 |
| Effisyn | 0.1 | 0.1 | 0.79 | 0.91 | 0.98 | 0.82 |
| CMO (Ours) | **0.86** | 0.85 | **1.00** | **1.00** | 0.99 | **0.96** |

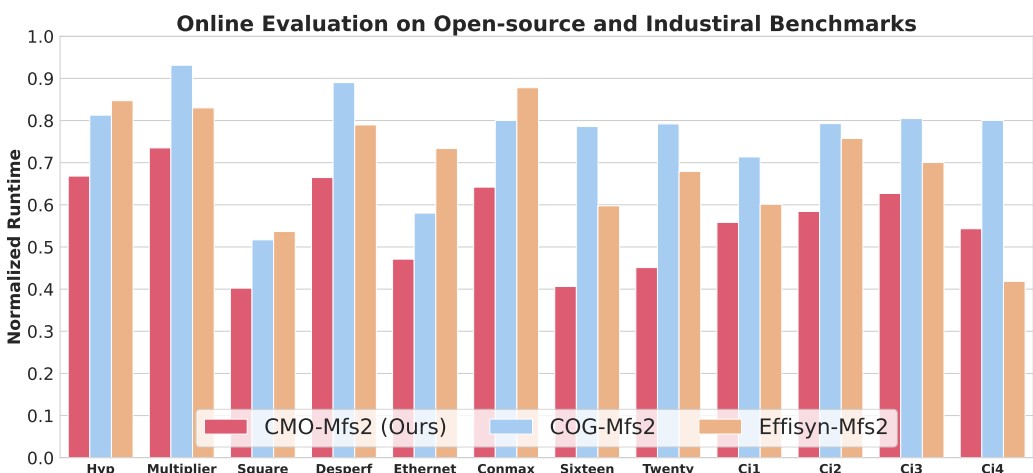

Figure 4: We compare our CMO with the COG and Effisyn on online runtime. The results demonstrate that our approach achieves significant runtime improvement with the baselines.

features. CMO is our proposed graph enhanced symbolic discovery framework. Moreover, we compare our CMO with the other five lightweight baselines. The baselines include two state-of-the-art (SOTA) searching-based SR approaches, two traditional lightweight ML models, and a Random approach. Please refer to Appendix E.2 for the implementation details of these baselines. **In the Online phase,** we evaluate the efficiency and QoR of CMO. Specifically, we present details as follows. (1) **Evaluation metrics** In terms of the efficiency of the heuristics, we use the runtime metric. In terms of QoR, we mainly use the optimized node, i.e., the number of the optimized circuit nodes, which has a significant impact on the final chip area. Moreover, we also use the depth (i.e., level) of the optimized circuits, which is a proxy metric for the delay of the designed chip. (2) **Evaluated methods** In the online phase, we introduce a new heuristic called X-Mfs2, which integrates the learned scoring function "X" into the default Mfs2 heuristic. In our experiments, "X" refers to the COG, Effisyn, the five lightweight baselines and our CMO.

**Evaluation Strategy** In real industrial scenarios, we hope that the trained model can generalize to many unseen circuits. Inspired by the leave-one-domain-out cross-validation strategy commonly used in previous literature (Wang et al., 2022a), we design twelve leave-one-out datasets for evaluation. Specifically, given a benchmark, we construct a dataset by setting one circuit as the testing dataset, and the other circuits as the training dataset. Please refer to D.2 for more details.

**Experiment 1. Comparitive Evaluation** In this subsection, we compare our CMO with state-of-the-art GPU-based (COG) and human-designed (Effisyn) methods in terms of generalization perfor-

Table 2: We compare the Default Mfs2 heuristic with our 2CMO-Mfs2 heuristic with the hyperparameter $k$ set as $30\%$, $40\%$ and $50\%$ on open-source and industrial circuits. Optimized Nd denotes the node number (size) of circuits, and Lev denotes the level (depth) of circuits. We define an Improvement metric by $\frac{M(\text{Default}) - M(\text{Ours})}{M(\text{Default})}$, where $M(\cdot)$ denotes the Nd, Lev, or Time.

| Open-source Circuits | Hyp | | | | Desperf | | | |
|---|---|---|---|---|---|---|---|---|
| Method | Lev ↓ | Improvement ↑ (Lev, %) | Time (s) ↓ | Improvement ↑ (Time, %) | Optimized Nd ↓ | Improvement ↑ (Optimized Nd, %) | Time (s) ↓ | Improvement ↑ (Time, %) |
| Default (Mfs2) | 8259.00 | NA | 319.33 | NA | 30853.00 | NA | 36.76 | NA |
| CMO-Mfs (0.5, Ours) | 8259.00 | 0 | 158.49 | 50.37 | 30910.00 | -0.18 | 26.40 | 28.19 |
| 2CMO-Mfs (0.3, Ours) | 5762.00 | 30.23 | 127.51 | 60.07 | 29392.00 | 2.31 | 30.16 | 17.96 |
| 2CMO-Mfs (0.4, Ours) | 5762.00 | 30.23 | 170.45 | 46.62 | 29175.67 | 2.50 | 38.20 | -3.92 |

| very Large-scale Circuits | Sixteen | | | | Twenty | | | |
|---|---|---|---|---|---|---|---|---|
| Method | Optimized Nd ↓ | Improvement ↑ (Optimized Nd, %) | Time (s) ↓ | Improvement ↑ (Time, %) | Optimized Nd ↓ | Improvement ↑ (Optimized Nd, %) | Time (s) ↓ | Improvement ↑ (Time, %) |
| Default (Mfs2) | 6017631.00 | NA | 78784.04 | NA | 7693089.00 | NA | 108735.49 | NA |
| CMO-Mfs2 (0.5, Ours) | 6018729.00 | -0.001 | 32001.27 | 59.38 | 7694455.00 | -0.002 | 56965.94 | 47.61 |
| 2CMO-Mfs2 (0.3, Ours) | 5434092.00 | 9.70 | 36425.15 | 53.77 | 6877483.00 | 10.60 | 59786.55 | 45.02 |
| 2CMO-Mfs2 (0.4, Ours) | 5433745.00 | 9.70 | 46572.91 | 40.80 | 6877158.00 | 10.61 | 75956.18 | 30.15 |

| Industrial Circuits | Ci1 | | | | Ci2 | | | |
|---|---|---|---|---|---|---|---|---|
| Method | Lev ↓ | Improvement ↑ (Lev, %) | Time (s) ↓ | Improvement ↑ (Time, %) | Optimized Nd ↓ | Improvement ↑ (Optimized Nd, %) | Time (s) ↓ | Improvement ↑ (Time, %) |
| Default (Mfs2) | 47.00 | NA | 180.35 | NA | 99245.00 | NA | 78.07 | NA |
| CMO-Mfs2 (0.5, Ours) | 47.00 | 0.00 | 113.08 | 37.30 | 99245.00 | 0.00 | 45.61 | 41.58 |
| 2CMO-Mfs2 (0.3, Ours) | 45.00 | 4.26 | 142.96 | 20.73 | 94240.00 | 5.04 | 64.37 | 17.55 |
| 2CMO-Mfs2 (0.4, Ours) | 45.00 | 4.26 | 177.04 | 1.84 | 93184.00 | 6.11 | 81.07 | -3.84 |

mance. The evaluation follows the proposed generalization strategy and is conducted on two widely used open-source benchmarks and one industrial benchmark. We use the offline prediction recall as the evaluation metrix. Results in Table 1 shows that our CMO outperforms the GNN on half of the circuits in terms of the prediction recall, demonstrating the strong generalization capability of our CMO. Moreover, our CMO achieves an average improvement of 36% in prediction recall when compared to Effisyn. Additionally, CMO achieves a prediction recall exceeding 85% across most test circuits, indicating it can maintain applying most of the effective transformations. In the online phase, we mainly focus on the efficiency of the X-Mfs2 heuristics. To ensure a fair comparison for efficiency, we maintain comparable optimization performance across all comparison methods. However, due to the significant prediction shift between Effisyn and our CMO as shown in Table 1, we have to adjust a higher $k$ for Effisyn to achieve comparable optimization performance (see Appendix C.3 for the optimization results). Thus, we maintain the top 50% accuracy for our CMO and COG and apply the top 70% for Effisyn. The results in Figure 4 show that our CMO-Mfs2 significantly outperforms the baselines in terms of efficiency. Specifically, our CMO achieves an average improvement of 21.05% and 15.07% over the COG and Effisyn on the open-source and industrial benchmarks. Overall, the offline and online results demonstrate that our CMO can not only achieve comparable prediction performance to the well-designed teacher GNN but also outperform all of the baselines in terms of heuristic efficiency. Due to limited space, we provide the online optimization results and more comparison results with other lightweight baselines in Appendix C.3.

**Experiment 2. Improving Efficiency and QoR of the LO heuristic** In this subsection, we conduct experiments on six challenging circuits to demonstrate that our method can not only reduce runtime but also improve QoR, such as the sizes and depths of optimized circuits. The size and depth are critical metrics in chip design, as they are proxies for the final area and delay of chips. We first show that our CMO could improve the efficiency of the Mfs2 heuristic with comparable optimization performance. Specifically, the results in Table 2 demonstrate that our CMO significantly achieves an average runtime improvement of 44.07% with marginal node degradation. In particular, our CMO achieves $2.5\times$ faster runtime on the very large-scale circuit Sixteen (about 13 hours). Then, we can sequentially apply CMO-Mfs2 multiple times rather than once to improve the QoR (i.e., 2CMO-Mfs2), as the runtime of CMO-Mfs2 is significantly shorter than that of Default Mfs2 heuristic. To achieve faster runtime, we set the hyperparameter $k$ as 30% and 40% rather than 50% to achieve faster runtime. Table 2 shows that 2CMO-Mfs2 significantly reduces the size and depth of optimized circuits while achieving faster runtime compared with the Default Mfs2 heuristic. Specifically, 2CMO-Mfs2 with $k = 40\%$ reduces the size/depth by $10.57\%$ on average while reducing the runtime by $18.60\%$ on the test circuits. Furthermore, suppose we want to achieve faster runtime in certain real-world scenarios, then we can set $k$ as a smaller value such as 30%. Table 2 shows that 2CMO-Mfs2 with $k = 30\%$ reduces the size/depth by $10.36\%$ on average with $35.85\%$ runtime reduction on the open-source and industrial circuits. In particular, our method achieves a significant reduction over the depth on Hyp, improving the level by $30.23\%$. Overall, the results suggest that

our efficient CMO-Mfs2 can significantly improve the QoR while achieving faster runtime, yielding a substantial economic value in chip design. Please refer to Appendix C.4 for more results.

**Experiment 3. Ablation Study** In this subsection, we conduct an ablation study to understand the individual contribution of each component of our CMO. To this end, we compare our CMO with its variants, i.e., CMO without GESD and CMO without SFD on two widely-used open-source benchmarks. The results in Table 3 suggest the following two conclusions. First, CMO without GESD significantly outperforms CMO without GESD and SFD in terms of offline prediction recall, demonstrating the impor-

Table 3: The ablation study results.

| Circuit | Multiplier | Square | Hyp |
|---|---|---|---|
| Method | Recall ↑ | Recall ↑ | Recall ↑ |
| CMO | **0.96** | **0.98** | **0.99** |
| CMO without GESD | 0.91 | 0.96 | 0.67 |
| CMO without SFD and GESD | 0.52 | 0.72 | 0.93 |
| Circuit | DesPerf | Ethernet | Conmax |
| Method | Recall ↑ | Recall ↑ | Recall ↑ |
| CMO | **0.80** | **0.72** | **0.84** |
| CMO without GESD | **0.80** | 0.44 | 0.76 |
| CMO without SFD and GESD | 0.60 | 0.42 | 0.45 |

tance of structural-semantic symbolic learning formulation. Second, CMO also significantly outperforms CMO without GESD. This demonstrates that GESD can effectively distill a high-performance symbolic function from the teacher GNN. We defer detailed results in Appendix C.5.

**Experiment 4. Strengths for Deployment** In this subsection, we conduct extensive experiments to demonstrate the appealing features of our CMO on high optimization performance, inference efficiency, and interpretability. Specifically, we present a detailed analysis as follows.

**Inference Efficiency** We compare the model inference time of our CMO with the GNN approach and human-designed approach on open-source circuits and industrial circuits. Results in Table 4 demonstrate that CMO learned scoring functions are extremely efficient on purely CPU-based machines, achieving a speedup of several hundred times compared to GNN. The efficiency comes from both the graph-independency inference and the concise sym-

Table 4: The model inference results show that our CMO are extremely efficient for inference compared to the other approaches when executed on CPU-based LO tools.

| | Hyp | DesPerf | Sixteen | Twenty | Ci1 | Ci2 |
|---|---|---|---|---|---|---|
| Method | Time(s) ↓ | Time(s) ↓ | Time(s) ↓ | Time(s) ↓ | Time(s) ↓ | Time(s) ↓ |
| COG | 28.28 | 6.24 | 1377.66 | 1787.07 | 30.00 | 18.23 |
| Effisyn | 0.15 | 0.06 | 9.67 | 12.19 | 0.48 | 0.22 |
| CMO (Ours) | **0.06** | **0.05** | **4.16** | **4.96** | **0.17** | **0.06** |

bolic functions. Moreover, while the human-designed scoring function demonstrates notable inference efficiency, our method achieves superior speed as we learn a simpler scoring function.

**Interpretability** Compared to GNN, CMO generates a concise symbolic function with better interpretability, enhancing simplicity and reliability for employment. We report the learned structural functions and boolean semantic functions in Table 16. We observed that $x_2$, which represents the node level, is approximately positively correlated with the node scores, indicating its significance in ineffective node prediction. Furthermore, some critical hyperparameters in the traditional human-designed scoring function heavily rely on the node level (Li et al., 2023). The intuition behind the design of this scoring function is highly similar to the learned symbolic policies in Table 16. Thus, we believe our CMO can help researchers further understand and design effective scoring functions.

## 6 CONCLUSION

To enable efficient Logic Optimization (LO), previous studies propose to use scoring functions to predict and prune ineffective nodes in LO heuristics. However, existing functions struggle to balance inference efficiency, interpretability, and generalization performance. To address this, we propose CMO, a novel Circuit Symbolic Learning Framework that learns efficient, interpretable, and high-performance symbolic functions. The key contribution of CMO is a graph-enhanced symbolic learning framework that distills dark knowledge from a well-designed GNN to improve the generalization of learned functions. Extensive experiments on three challenging benchmarks show that the interpretable learned functions outperform previous SOTA GPU-based and human-designed methods in terms of efficiency and generalization. Moreover, CMO significantly improves the Mfs2 heuristic's efficiency with comparable optimization performance on a CPU-based machine, achieving up to 2.5× faster runtime.

## REPRODUCIBILITY STATEMENT

In this study, to ensure the reproducibility of our approach, we provide key information from the main text and Appendix as follows.

1. **Algorithm.** We provide the architecture and pseudocode of our approach in Section 4 and Appendix E.5. Moreover, we will make our source code publicly available once the paper is accepted for publication.

2. **Source Code.** To facilitate the evaluation process and support a thorough review, we have released our source code at the following link: `https://gitee.com/yinqi-bai/cmo.git`.

3. **Experimental Details.** We provide experiment settings in Section 5 and Appendix E.1.

## 7 ACKNOWLEDGEMENTS

The authors would like to thank all the anonymous reviewers for their insightful comments and valuable suggestions. This work was supported by the National Key R&D Program of China under contract 2022ZD0119801 and the National Nature Science Foundations of China grants U23A20388, 62021001 and 623B1022.

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

## A    RELATED WORK

**Learning-based Methods for Symbolic Discovery** Several recent approaches utilize deep learning for symbolic discovery. These methods generally fall into three categories—Evolutionary, Pre-trained, and Searching-based. Genetic programming (GP) (O'Neill, 2009) is one of the classic evolutionary symbolic regression methods, operating by maintaining a population of expression "individuals" that evolve using evolutionary operations such as selection, crossover, and mutation. While GP can be effective, it tends to struggle with scalability for larger problems. In recent years, pre-trained symbolic regression methods have shown advantages in fast inference and have successfully discovered large input (with up to twelve) symbolic functions (d'Ascoli et al., 2023; Biggio et al., 2021; Kamienny et al., 2022). However, these methods are limited by high training costs and data generalization challenges. Unlike the methods mentioned above, the searching-based method explores the discrete symbolic operator space and finds the best function that maximizes the fitness of the given dataset. The mainstream symbolic regression frameworks are based on Monte Carlo

tree search (Sun et al., 2023; Xu et al., 2024) and sequence prediction models such as RNN (Petersen et al., 2020). These methods have achieved state-of-the-art performance on multiple benchmarks.

**Symbolic Distillation from the Trained GNNs** Motivated by the high expressive capacity but opaque nature of GNNs, previous studies have sought to distill interpretable symbolic functions from GNNs to approximate their mapping functions. First, (Cranmer et al., 2020a) introduces a framework that distills interpretable symbolic functions from trained GNNs for scientific discovery. This approach initially trains the neural network models and then utilizes a symbolic learning model to iteratively approximate both the message-passing and aggregation functions with symbolic representations. However, (Kuang et al.) found that distilling trained GNNs layer by layer often results in suboptimal performance. To address this, they propose learning interpretable symbolic policies directly from the general bipartite graph representation in an end-to-end manner. Nonetheless, the symbolic functions produced in this way can become complex, as they must map the relationships between graph inputs and outputs. In contrast, our work focuses on using only the node features as input to enhance inference efficiency and ensure better interpretability.

**Machine Learning for Combinatorial Optimization** Logic optimization, a critical component of logic synthesis in electronic design automation (EDA), fundamentally constitutes a combinatorial optimization problem. In recent years, the application of machine learning techniques to address combinatorial optimization challenges has emerged as a prominent and actively researched area of interest, particularly in the EDA domain (Korte et al., 2011; Cook et al., 1994; Wang et al., 2023a; 2024a; 2022b; 2023b; Li et al., 2024; Ling et al., 2024; Geng et al., 2023a; 2025a;b). This paradigm shift is driven by the increasing complexity of modern integrated circuits and the limitations of traditional heuristic-based approaches. Machine learning offers promising solutions through its ability to learn complex patterns from large-scale design data, enabling more efficient exploration of the solution space and potentially leading to better quality-of-results (QoR) in terms of power, performance, and area (PPA) optimization. Recent advancements in deep reinforcement learning and graph neural networks have shown particular promise in tackling the intricate dependencies and constraints inherent in logic optimization problems (Geng et al., 2023b).

## B MORE DETAILS ON THE BACKGROUND

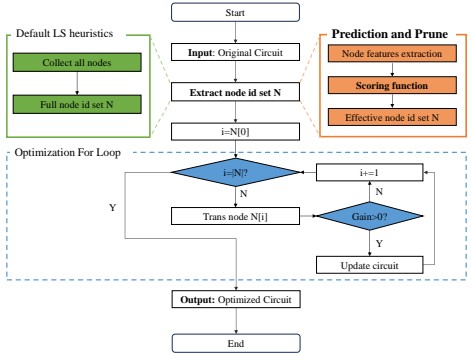

Figure 5: The Prediction and Prune framework.

**Logic Optimization (LO)** Driven by Moore's law, the complexity of chip design has grown exponentially (Khailany, 2020; Lopera et al., 2021; Huang et al., 2021; Mirhoseini et al., 2021; Ren & Hu, 2023; Wang et al., 2025; 2024c). To manage this complexity, chip design workflows have incorporated multiple Electronic Design Automation (EDA) tools to synthesize, simulate, test, and verify different circuit designs efficiently and reliably. Among these tools, A LO tool–which aims to optimize the circuit represented by a Boolean network–is one of the most important modules in the EDA tools. LO typically involves two stages: pre-mapping optimization and post-mapping optimization (Hosny et al., 2020; Ren & Hu, 2023; Wang et al., 2024d). During the pre-mapping optimization phase, LO heuristics such as Rewrite (Bertacco et al., 1997), Resub (Brayton, 2006), and Refactor (Brayton, 1982) are used to improve the input circuit. Following this, in the technology mapping phase, the optimized logic circuit is mapped to the available technology library, e.g., a standard-cell netlist (Brayton & Kam) or k-input lookup tables (Mishchenko et al., 2007). Finally,

the post-mapping optimization phase applies LO heuristics like Mfs2 (Mishchenko et al., 2011) to further refine and enhance the mapped circuit.

**Logic Optimization heuristics** To tackle the LO task, many researchers have developed a rich set of LO heuristics. For instance, researchers have developed Rewrite (Bertacco et al., 1997) and Resub (Brayton, 2006) for pre-mapping optimization, while Mfs2 (Mishchenko et al., 2011) is designed for post-mapping optimization. These LO heuristics follow the paradigm as shown in Figure 5. Specifically, these heuristics traverse the Boolean network in a topological order from PIs to POs and apply transformations to subgraphs rooted at each node sequentially for all nodes. However, previous literature(Wang et al.) found that these heuristics can be highly time-consuming due to a large number of ineffective transformations. To address this problem, we follow the new heuristics paradigm proposed by (Wang et al.) that can significantly improve the efficiency of LO heuristics by learning a classifier to predict nodes with ineffective transformations and avoid applying transformations on these nodes. In this paper, we focus on optimizing the post-mapping operator Mfs2 (Mishchenko et al., 2011), which stands out as the most time-consuming one among all commonly used LO heuristics.

**Circuit Representation** In the LO stage, a circuit is usually modeled by a Boolean network (Wang et al., 2024b). In this paper, we use the terms Boolean network and circuit interchangeably. A Boolean network is a directed acyclic graph (DAG), where nodes correspond to Boolean functions and directed edges correspond to wires connecting these functions. A Boolean function takes the form $f : \mathbf{B}^n \to \mathbf{B}$, where $\mathbf{B} = \{0, 1\}$ denotes the Boolean domain. Given a node, its fanins are nodes connected by incoming edges of this node, and its fanouts are nodes connected by outgoing edges of this node. The primary inputs (PIs) are nodes with no fanin, and the primary outputs (POs) are nodes with no fanout. The *size* of a circuit denotes the number of nodes in the DAG. The depth (level) of a circuit denotes the maximal length of a path from a PI to a PO in the DAG. The size and depth of a circuit are proxy metrics for the area and delay of the circuit, respectively.

**Symbolic Expression Tree for Learning Symbolic Functions** Given a dataset $\mathcal{D}$, symbolic learning approaches aim to learn a symbolic function $f$ that best fits the dataset. To this end, symbolic functions are often represented by an algebraic expression tree, where internal nodes are operators and terminal nodes are input variables and/or constants (Sun et al., 2023; Liu et al., 2024b; Kuang et al., 2024; Liu et al., 2025; 2024a). We assume $\tau = [\tau_1, \ldots, \tau_n]$ is a pre-order traversal of such an expression tree. Each $\tau_i$ is an operator, input variable, or constant selected from a library of possible tokens and there is a one-to-one correspondence between an expression tree and its pre-order traversal. An expression tree is also used to express mathematical functions with operators such as $[+, -, \times, \div, \sin]$.

**Graph Neural Network (GNN)** Recently, many researchers have achieved remarkable success in applying Graph Neural Networks (GNNs) to extract abundant node information from circuit graphs (Wu et al., 2023; Shi et al., 2025; 2024). As far as we are aware, (Wang et al.) applies GNN to predict nodes with ineffective transformations for the LO task. The input of GNN is a heuristic-designed subgraph rooted as the node, and the output is a continuous score ranging from 0 to 1 for the node. Based on the scores, nodes in circuits are ranked and the top k nodes are selected to be positive. However, despite achieving a high accuracy on ineffective node prediction, GNN heavily relies on a GPU for efficient inference while many EDA developers may not have access to high-end GPUs, which thus severely limits its application into purely CPU-based LO tools.

**Graph-based Knowledge Distillation** Knowledge Distillation (KD) aims to transfer knowledge from a large, complex teacher model to a smaller, more efficient student model (Gou et al., 2021). The main idea is that the student model mimics the teacher model in order to obtain a competitive or even superior performance. Recently, Graph-based Knowledge Distillation has emerged as a significant research direction within KD research. This process involves distilling knowledge from a Graph Neural Network (GNN), into a more lightweight student model, such as a Multilayer Perceptron (MLP) (Zhang et al., 2022; Tian et al., 2022) or symbolic functions (Cranmer et al., 2020b; Kuang et al.), which only relies on node features as input. By eliminating the graph dependency during inference, this approach enables a more efficient and faster student model while retaining the high predictive performance learned by the GNN.

## C  ADDITIONAL RESULTS

### C.1  MORE MOTIVATING RESULTS

**Existence of high-performance symbolic functions** To verify whether concise and high-performance symbolic functions exist in our computational synthesis (LO) task, we apply the symbolic tree search algorithm to all valid circuits—specifically, those with effective nodes—in the EPFL benchmark. In this evaluation, we utilize the top 50% accuracy metric as the simulated rewards to gauge performance. As shown in Figure 6, an impressive 93.3% of the circuits (14 out of 15) achieve a top 50% accuracy exceeding 99%. Furthermore, our analysis indicates that all learned symbolic functions are not only short but also remarkably concise. This observation suggests a strong potential for leveraging symbolic learning approaches in our LO task, indicating that such methods could efficiently enhance performance while maintaining clarity and simplicity in the symbolic representations.

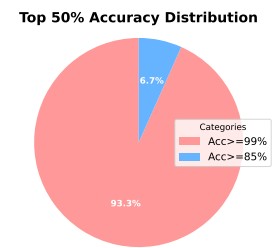

Figure 6: High-performance symbolic functions exist in the open-source benchmarks.

### C.2  THE IMPORTANCE OF THE PREDICTION RECALL ON OPTIMIZATION PERFORMANCE

To analyze the relationship between the prediction recall of effective nodes and the optimization performance of heuristics, we evaluate the optimization performance of the Random method with different values of the hyperparameter $k$. Note that Random is a baseline that randomly predicts a score between $[0, 1]$ for each node, and selects the top $k$ nodes to apply node-level transformations. Specifically, we report the recall and optimization performance (i.e., And Reduction) of Random with different values of $k$ in Table 5. The results indicate that the value of $k$ is approximately linearly positively correlated with the recall, and the recall is approximately linearly positively correlated with the optimization performance as well. Therefore, to maintain the optimization performance of heuristics, the prediction recall of our model should be as high as possible.

### C.3  MORE RESULTS FOR COMPARATIVE EVALUATION

Due to the significant Top 50% accuracy shift between Effisyn and our CMO shown in Table 1, we have to adjust a higher $k$ for the Effisyn to achieve comparable prediction recall and thus comparable online optimization performance (shown in Table 5). Specifically, we choose $k$ as 70% for the Effisyn as too large $k$ will bring higher time cost. Results shown in Table 6 demonstrate that our CMO achieves an average improvement of 21.05% compared to COG while maintaining comparable optimization performance (normalized And reduction is 0.92 versus 0.91). Additionally, when compared to Effisyn, our CMO shows an average improvement of 15.07% while preserving similar optimization performance across most circuits. Note that Effisyn exhibits very low optimization performance on two very large-scale circuits, even with $k$ set to 70%. For these circuits, $k$ may need to be adjusted to 90% or higher, which would result in significantly increased time costs. Moreover, we compare our CMO with the other five lightweight baselines on open-source circuits. The results in Tables 8 and 9 demonstrate that our CMO significantly outperforms these baselines in both generalization performance and efficiency.

### C.4  MORE IMPROVING EFFICIENCY AND QOR RESULTS

In this subsection, we provide more results about the improving efficiency and QoR results in Table 18. We first show that our CMO could improve the efficiency of the Mfs2 heuristic with comparable optimization performance. Specifically, the results in Table 2 demonstrate that our CMO significantly achieves an average runtime improvement of 44.54% with marginal node degradation. In particular, our CMO achieves 2.5× faster runtime on the very large-scale circuit Sixteen (about 13 hours). Then, we can sequentially apply CMO-Mfs2 multiple times rather than once to improve the QoR (i.e., 2CMO-Mfs2), as the runtime of CMO-Mfs2 is significantly shorter than that of Default Mfs2 heuristic. To achieve faster runtime, we set the hyperparameter $k$ as 30% and 40% rather than

$50\%$ to achieve faster runtime. Table 2 shows that 2CMO-Mfs2 significantly reduces the size and depth of optimized circuits while achieving faster runtime compared with the Default Mfs2 heuristic. Specifically, 2CMO-Mfs2 with $k = 40\%$ reduces the size/depth by $5.82\%$ on average while reducing the runtime by $7.99\%$ on the test circuits. Furthermore, suppose we want to achieve faster runtime in certain real-world scenarios, then we can set $k$ as a smaller value such as $30\%$. Table 2 shows that 2CMO-Mfs2 with $k = 30\%$ reduces the size/depth by $5.60\%$ on average with $26.16\%$ runtime reduction on the open-source and industrial circuits. In particular, our method achieves a significant reduction over the depth on Hyp, improving the level by $30.23\%$. Overall, the results suggest that our efficient CMO-Mfs2 can significantly improve the QoR while achieving faster runtime, yielding a substantial economic value in chip design.

### C.5 MORE ABLATION STUDY RESULTS

In this subsection, we provide more ablation combinations and corresponding results. Specifically, CMO without structural-semantic feature decomposition denotes that we apply our graph distillation module directly to the original high-dimensional circuit data. The results in Table 7 show that CMO without SFD outperforms CMO without SFD and GSD in terms of the offline prediction recall, indicating the robustness and ability of our graph distillation module.

### C.6 MORE RESULTS OF STRENGTHS FOR DEPLOYMENT

In this subsection, we provide more results of our method's deployment strengths in terms of optimization performance and inference efficiency.

**High Optimization Performance** There are many lightweight and interpretable models suitable for industrial deployment. Compared to these models, our symbolic discovery framework take advantage in achieving higher optimization performance. We compare our CMO with five lightweight baselines on all of the test circuits. The results in Table 8 demonstrate that our method significantly outperforms all of the baselines in prediction recall. Specifically, it achieves an average improvement of 24% and 21% to the state-of-the-art symbolic discovery approach SPL and a powerful traditional machine learning approach XGBoost.

**More results for inference efficiency** We provide detailed model inference time results for all test circuits. Table 10 demonstrates that the runtime of the GNN model for COG is hundreds of times greater than that of CMO when executed on purely CPU-based machines. Additionally, results in Table 11 show that the inference time of the GNN model accounts for up to 29% of the total end-to-end heuristic execution time, which significantly impacts the efficiency of LO heuristics. Therefore, while the graph-based machine learning model achieves optimization performance comparable to that of CMO, these methods are limited to deployment in modern CPU-based LO tools due to the high inference costs.

## D DETAILS OF DATASETS USED IN THIS PAPER

### D.1 DESCRIPTION OF THREE USED BENCHMARKS

We provide detailed statistics of the circuits from EPFL and IWLS in Tables 12 and 13, respectively. Additionally, statistics for the very large-scale circuits and industrial circuits are shown in Tables 14 and 15. The industrial circuits consist of 27 circuits. In general, nodes correspond to logic gates and edges represent the wires connecting them. The fanins of a node are the nodes supplying input to it, while the fanouts are the nodes that it drives. Primary inputs (PIs) are nodes without fanins, and primary outputs (POs) are a subset of the network's nodes. Latches are specialized nodes in sequential circuits, and cubes represent subsets of input variables in Boolean functions. Lev refers to the depth of the DAG, measured by the maximum number of edges between PIs and POs.

### D.2 DATASETS FOR EVALUATION ON OPEN-SOURCE BENCHMARKS

For each circuit and a given X heuristic, we collect the circuit dataset by applying the X heuristic to optimize the circuit and collecting the node features $\{\mathbf{x}_i\}_{i=1}^n$ and labels $\{y_i\}_{i=1}^n$. We found that

there are a small number of circuits with no effective nodes. We discard these circuits, as we can directly avoid applying transformations to these circuits without learning a model.

Specifically, using the leave-one-out evaluation strategy with the EPFL benchmark, we construct three datasets for evaluation. One of the three circuit datasets—collected from Hyp, Multiplier, and Square—serves as the testing dataset, while the remaining datasets are used for training. Similarly, under the leave-one-out evaluation strategy with the IWLS benchmark, we create three datasets, selecting one of the four circuit datasets from Ethernet, Conmax, and Desperf as the testing dataset and using the others for training. To foster the machine learning community in Logic Optimization, we will release the datasets once the paper is accepted for publication.

### D.3 DATASETS FOR EVALUATION ON INDUSTRIAL CIRCUITS AND VERY LARGE-SCALE CIRCUITS

In terms of the industrial circuits, we report a statistical description of the training and testing circuits in Table 15. As shown in Tables 15, 12, and 13, the industrial circuits consist of 27 industrial circuits, where the circuit sizes range from $2,775$ to $788,288$, which are much larger in size than open-source circuits. Using the leave-one-out evaluation strategy, we evaluate our method with 23 circuits for training and 4 circuits for testing.

For very large-scale circuits (up to twenty million nodes), we provide a detailed description in Table 14. Given the limited number of very large-scale circuits, we utilize circuits from the EPFL dataset outlined in Table 12 along with two additional very large-scale circuits to construct our datasets. Specifically, we adopt the following approach: (1) The Sixteen circuit is designated as the testing dataset, while the remaining circuits serve as the training dataset. (2) The Twenty circuit is set as the testing dataset, with the rest used for training. To support the machine learning community in Logic Optimization, we will release these datasets upon acceptance of the paper for publication.

### D.4 VISUALIZATION OF THE CIRCUIT GRAPH

In the LO stage, a circuit is usually modeled by a DAG. Common types of DAGs for LO include And-Inverter Graphs (AIGs) for pre-mapping optimization and K-Input Look-Up Tables (K-LUTs) for post-mapping optimization. In the pre-mapping optimization phase, an AIG is a DAG containing four types of nodes: the constant, PIs, POs, and two-input And (And2) nodes. A graph edge is either complemented or not. A complemented edge indicates that the signal is complemented. In the post-mapping optimization phase, a K-LUT is a DAG with nodes corresponding to Look-Up Tables and directed edges corresponding to wires. A Look-Up Table in a K-LUT is a digital memory that implements the Boolean function of the node. To further illustrate the circuit graph, we visualize the AIG, K-LUT look-up table, and the circuit optimization process of a small circuit selected from IWLS2020 (Rai et al., 2021) in Figure 7.

## E DETAILS OF METHODS AND EXPERIMENTAL SETTINGS

### E.1 DETAILS OF EXPERIMENTAL SETUP

#### E.1.1 OPTIMIZATION SEQUENCE FLOWS

**Optimization Sequence Flows for Collecting Data and Evaluation** In the industrial setting, we usually apply a sequence of Logic Optimization (LO) heuristics to optimize an input circuit. Thus, we follow the setting throughout all experiments unless mentioned otherwise. Specifically, in terms of the Mfs2 heuristic, we apply the sequence of heuristics, i.e., *strash; dch; if -C 12; mfs2 -W 4 -M 5000*, to collect data and evaluate the performance of the Default Mfs2 heuristic and our CMO. Note that the optimization sequence flow is a standard academic flow for evaluating the Default Mfs2 heuristic, which follows previous work (Mishchenko et al., 2011).

**Optimization Sequence Flows for Evaluating 2CMO-Mfs2** To apply our CMO twice, we apply the sequence of heuristics, i.e., *strash; dch; if -C 12; mfs2 -W 4 -M 5000; strash; if -C 12; mfs2 -W 4 -M 5000*, to evaluate the performance of 2CMO. Note that the mfs2 heuristic is a post-mapping optimization heuristic, whose input DAG is a k-input look-up table graph (K-LUTs). Moreover, the strash heuristic transforms the current circuit representation into an And-Inverter Graph (AIG) by

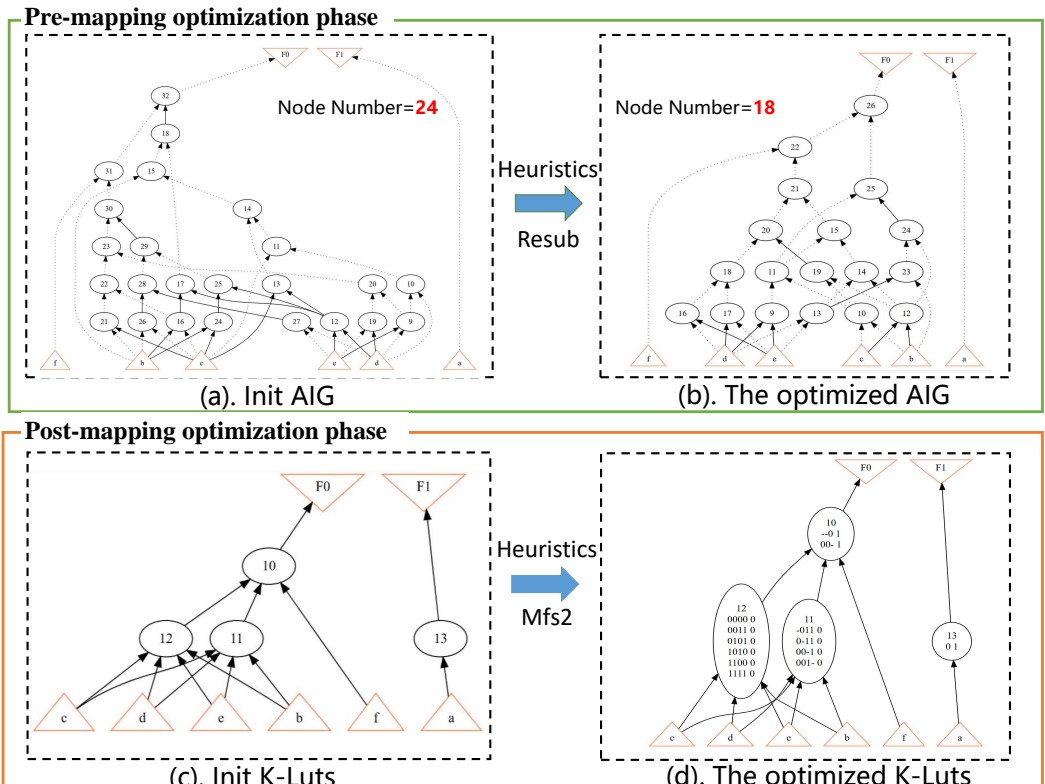

Figure 7: Visualization of the And-Inverter Graph (AIG) and K-Input Look-Up Tables (K-LUTs). In the pre-mapping circuit optimization phase, LO heuristics such as resub, are applied to optimize the AIG. In the post-mapping phase, LO heuristics such as mfs2, are used to optimize the K-LUTs.

one-level structural hashing. Then, the *if* (Mishchenko et al., 2007) heuristic maps an AIG into a K-LUTs. Finally, the Mfs2 heuristic optimizes the input K-LUTs.

### E.1.2 TOP K ACCURACY METRIC

The ineffective Node-Level Transformations problem in many LO heuristics——that is, the number of effective nodes is far fewer than ineffective nodes——leads to an extreme class imbalance in the training dataset. Therefore, the learned classifiers always suffer from negative bias and thus 0.5 is an inappropriate threshold for evaluating whether a sample is positive. To tackle this problem, we follow (Wang et al.) to view the prediction task as a scoring function. Specifically, we sort all the nodes according to the prediction scores given by our learned symbolic functions and select the top $k$ nodes. That is, the top $k$ nodes are predicted positive, and the other nodes are predicted negative. Then, the top $k$ accuracy metric is defined by the recall, i.e., the fraction of true positive nodes that are predicted to be positive.

### E.1.3 IMPLEMENTATION OF THE TRAINING DETAILS

In our implementation, we configured the weight factor $\lambda$ between $L_{teacher}$ and $L_{student}$ as 0.5 to balance the objective functions. We averaged the results over three different random seeds during training. Additionally, a regularization factor $\rho = 0.99$ was employed to penalize excessive expression length, encouraging more compact and interpretable solutions. The training process utilized Monte Carlo Tree Search (MCTS), running 10,000 episodes per iteration across 20 iterations. After training, the best-performing model on the training set from all 20 iterations was selected, ensuring both stability and optimal performance. After discovering structural and semantic functions from the training dataset, we fuse their scores by setting the $w$ as the median of the structural features' outputs.

E.2 IMPLEMENTATION OF THE BASELINES

In this part, we present a detailed description of all the baselines used in this paper.

**COG.** COG is a well-designed 2-layer graph convolutional neural network that can achieve high optimization performance (Wang et al.). Specifically, it constructs a bipartite graph as input and learns domain-invariant representation to achieve high generalization capability.

**Effisyn**. Effisyn is a human-designed nonlinear symbolic function (Li et al., 2023). Specifically, in human-designed symbolic scoring functions, experts manually design the structure of the function and extract key parameters from training circuit data to form a complete symbolic scoring function. This process involves identifying relevant characteristics of the circuit and carefully selecting or engineering the symbolic terms that best capture the underlying behavior of the system. However, designing and developing these functions is extremely challenging as it requires extensive expert knowledge and manual tuning.

**SPL.** SPL (Sun et al., 2023) is a search-based symbolic regression method that employs a Monte Carlo tree search (MCTS) agent to explore optimal expression trees based on measurement data. SPL is one of the SOTA symbolic learning methods.

**DSR.** DSR (Petersen et al., 2020) is a leading deep learning-based symbolic learning method that employs a gradient-based risk-seeking RL approach combined with a recurrent neural network (RNN) to generate a probability distribution over expressions.

**RidgeLR.** RidgeLR (Hoerl & Kennard, 1970) is a regularized version of linear regression that adds an L2 penalty to the loss function to prevent overfitting. This regularization term helps reduce the model's sensitivity to multicollinearity and large coefficients by shrinking them, making RidgeLR effective for regression tasks, especially when dealing with high-dimensional or highly correlated features. It is commonly used in regression problems where overfitting needs to be controlled.

**XGBoost.** XGBoost (Chen & Guestrin, 2016) is a machine learning algorithm that builds an ensemble of decision trees using gradient boosting, optimizing accuracy by iteratively correcting errors from previous trees. It includes regularization to prevent overfitting and is highly efficient, making it well-suited for both regression and classification problems, especially with large-scale and complex datasets.

**Random.** Random is a baseline that randomly predicts a score between $[0, 1]$ for each node, and selects the top k nodes as positive samples to apply node-level transformations.

E.3 IMPLEMENTATION OF THE SEMANTIC FUNCTION LEARNING PROCESS

Once decomposing the init feature into structural and semantic components, we collect structural data $\mathcal{D}_{str} = \{\mathbf{x}_i^{str}, y_i\}_{i=1}^N$ and semantic data $\mathcal{D}_{sem} = \{\mathbf{x}_i^{sem}, y_i\}_{i=1}^N$, where $\mathbf{x}_i^{str}$ refers to structural node feature and $\mathbf{x}_i^{sem}$ refers to semantic node feature. To capture structural information, we model it as a mathematical symbolic learning problem, as the values of structural features can be approximated as continuous data, making them suitable for continuous mathematical symbolic regression. In contrast, learning mathematical functions for semantic information is challenging due to the discrete and binary nature of both feature values and labels. Thus, we formulate the semantic function as a Boolean symbolic learning problem, i.e., learning a boolean function $f^{sem} : \mathbf{B}^d \to \mathbf{B}$ that can accurately identify the effective nodes, where $\mathbf{B} = \{0, 1\}$ denotes the boolean feature domain. Given the binary outputs of Boolean functions, we can select 0.5 as the classification threshold and model it directly as a classification problem. This significantly reduces the difficulty of symbolic function learning compared to treating it as a function regression problem.

we have discussed the details of the structural function learning process in Section 4.2. Here we give a detailed implementation of the semantic boolean function learning process. Specifically, the semantic function follows the same symbolic tree search paradigm as the structural function learning. However, the difference mainly lies in the symbolic operator and designed reward function. The symbolic operators include three basic logic operators $and, or, not$, which are always used to represent the mapping from binary values to binary values. Moreover, due to the binary nature of the boolean function's output, we can model it as a classification problem and use 0.5 as the

classification threshold. The loss function designed for the boolean function learning is:

$$L = \frac{TP}{FP + TP} - \lambda * FP = \frac{\sum_{i=1}^{N} 1(y_i = 1 \wedge p_i \geq 0.5)}{\sum_{i=1}^{N} 1(y_i = 1)} - \lambda \sum_{i=1}^{N} 1(p_i \geq 0.5) \quad (3)$$

The reason for the loss function is to let the scoring model identify more effective nodes. The term $\lambda * FP$ is used as the regularization to prevent the boolean function predicts all of the nodes as positive. Once the boolean function accurately identifies all of the effective nodes, it could significantly improve the predictive performance of the final fusion symbolic functions.

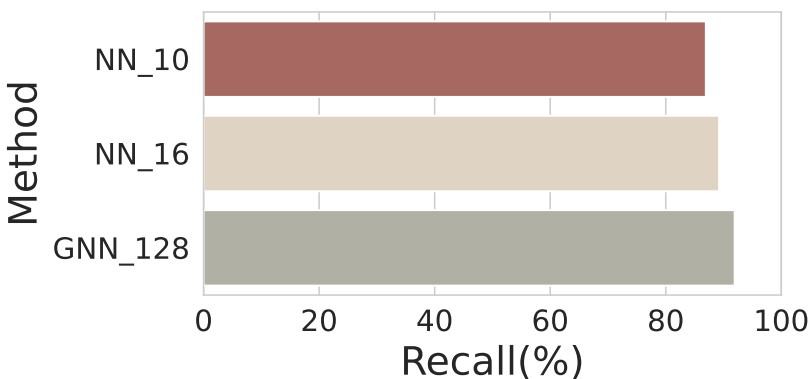

Figure 8: We use two simple multilayer perceptrons (MLP) to fit the relationship between the node features and the GNN's output. The results demonstrate that the MLP achieves comparable generalization performance to GNN.

### E.4 MORE DETAILS ABOUT THE NODE FEATURE DESIGNED

As shown in Table 17, the designed features for each node contain semantic information and structural information. Specifically, for the Mfs2 heuristic, the semantic features are designed with the truth table of the node (i.e., a 6-input look-up table) (Wang et al.). Moreover, the structural feature is a five-dimension vector that contains structural information of the circuit such as fanin/fanout number and level of the node. Overall, the node feature for the Mfs2 heuristic is a 69-dimensional vector.

### E.5 MORE DETAILS ABOUT THE GRAPH DISTILLATION APPROACH

We use equation 2 to evaluate every expression tree. Specifically, we use mean squared error and focal loss for the $L_{\text{teacher}}$ and $L_{\text{student}}$. That is,

$$L_{\text{teacher}} = \frac{1}{N} \sum_{i=1}^{N} (z_i - \hat{y}_i)^2 \quad (4)$$

and

$$L_{\text{student}} = -\alpha y (1 - \hat{y})^\gamma \log(\hat{y}) - (1 - \alpha)\hat{y}^\gamma \log(1 - \hat{y}),$$

$\hat{y}$ denotes the prediction output, $z$ denotes the soft output of GNN, and $y$ is the true labels. The motivation for the loss type of $L_{\text{teacher}}$ mainly comes from an observation that there exists a simple nonlinear mapping relationship from the circuit feature to the output of GNN that achieves comparable generalization performance. Specifically, we found that we can use a simple MLP to greatly fit the mapping relationship from the circuit feature to the output of GNN as shown in Figure 8.

Thus, we use MSE to directly learn information from the GNN outputs. Moreover, note that the number of effective nodes is far fewer than ineffective nodes, which poses a substantial challenge to the classification task. Therefore, we leverage the focal loss as the $L_{teacher}$, which has been shown successful in addressing the class imbalance for object detection tasks.

**Discussions on advantages** From the perspective of distillation, our approach allows for the distillation of GNNs into simplified symbolic functions. This represents a more thorough form of distillation compared to the conventional transformation from GNNs to MLPs. By reducing GNNs to symbolic functions, we not only preserve the essential predictive capabilities but also enhance interpretability and reduce computational complexity. From the viewpoint of symbolic learning, our symbolic search method leverages the design of fitness measures to discover symbolic functions with superior generalization capabilities. Unlike traditional machine learning models, which often struggle with overfitting and lack of interpretability, symbolic functions are derived through our method. offer a robust alternative. These functions are designed to effectively capture the underlying patterns in data, leading to better generalization in various applications.

---

**Algorithm 1** The training process

---

**Input:** Training dataset $\mathcal{D} = \{\mathbf{x}_i, y_i\}_{i=1}^N$;
**Output:** Semantic function $f^{sem}$, Structural function $f^{str}$;
   **Step 1:** Separate the initial dataset $\mathcal{D}$ into structural data $\mathcal{D}_{str} = \{\mathbf{x}_i^{str}, y_i\}_{i=1}^N$ and semantic data $\mathcal{D}_{sem} = \{\mathbf{x}_i^{sem}, y_i\}_{i=1}^N$;
   **Step 2:** Apply GESD (Algorithm 2) to learn the structural function $f^{str}$ from $\mathcal{D}_{str}$;
   **Step 3:** Apply GESD (Algorithm 2) to learn the semantic function $f^{sem}$ from $\mathcal{D}_{sem}$;

---

**Algorithm 2** GESD

---

**Input:** Training dataset $\mathcal{D} = \{\mathbf{x}_i, y_i\}_{i=1}^N$, Trained GNN model $f_{GNN}$;
**Output:** Optimal function $f^*$;
1: **for** each episode **do do**
2:    **Selection:** Initialize $s_0 = \emptyset$, $t = 0$;
3:    **while** $s_t$ is expandable and $t < t_{\max}$ **do**
4:       Choose $a_{t+1} = \arg\max_a \mathrm{UCT}(s_t, a)$;
5:       Take action $a_{t+1}$, observe $s'$, $NT$;
6:       $s_{t+1} \leftarrow s'$; mark as visited, $t \leftarrow t + 1$;
7:    **end while**
8:    **Expansion:** Randomly take an unvisited path with action $a$, observe $s'$;
9:    $s_{t+1} \leftarrow s'$; mark as visited, $t \leftarrow t + 1$;
10:    **Simulation:** Fix the starting point $s_t$;
11:    **for** each simulation **do do**
12:       **while** $s_t$ is non-terminal and $t < t_{\max}$ **do**
13:          Randomly take an action $a$, observe $s'$;
14:          $s_{t+1} \leftarrow s'$; $t \leftarrow t + 1$;
15:       **end while**
16:       **if** $s_{t+1}$ forms a complete expression tree **then**
17:          Project $f$;
18:          Calculate the function prediction $\hat{\mathbf{y}} = f(\mathbf{x})$;
19:          Calculate the GNN prediction $\mathbf{z} = f_{GNN}(\mathbf{x})$;
20:          Calculate the simulation reward:

$$r_{t+1} = \frac{\eta^n}{1 - \lambda L_{\text{label}}(\hat{\mathbf{y}}, \mathbf{y}) - (1 - \lambda) L_{\text{teacher}}(\hat{\mathbf{y}}, \mathbf{z})}$$

21:       **end if**
22:    **end for**
23:    **Backpropagate** the maximum simulation results and visited count;
24: **end for**

---

**Algorithm 3** The Inference process

---

**Input:** Test dataset $\mathcal{D}_{test} = \{\mathbf{x}_i, y_i\}_{i=1}^N$, structural function $f^{str}$, semantic function $f^{sem}$;
**Output:** The scores $\mathbf{s}$ for all nodes in $\mathcal{D}_{test}$;
   **step 1**: Separate the test dataset $\mathcal{D}_{test}$ into structural data $\mathcal{D}_{str} = \{\mathbf{x}_i^{str}, y_i\}_{i=1}^N$ and semantic data $\mathcal{D}_{sem} = \{\mathbf{x}_i^{sem}, y_i\}_{i=1}^N$;
   **step 2**: Calculate the structural scores $\mathbf{s}^{str}$ with $\mathbf{s}_i^{str} = f^{str}(\mathbf{x}_i^{str})$ and semantic scores $\mathbf{s}^{sem}$ with $\mathbf{s}_i^{sem} = f^{sem}(\mathbf{x}_i^{sem})$;
   **step 3**: Calculate the weight $w = Median(\mathbf{s}^{str})$;
   **step 4**: Calculate the final scores $\mathbf{s} = \mathbf{s}^{str} + w * \mathbf{s}^{sem}$ for all nodes in $\mathcal{D}_{test}$;

---

Table 5: We report the recall and optimization performance of the Mfs2 heuristic incorporated with Random models. And Reduction denotes the reduced number of nodes, i.e., optimization performance. Percent denotes the hyperparameter k, i.e., the percent of nodes to apply transformations.

| Hyp | | | Multiplier | | |
|---|---|---|---|---|---|
| Percent | Recall | And Reduction (AR) | Percent | Recall | And Reduction (AR) |
| 0.10 | 0.11 | 33.33 | 0.10 | 0.10 | 3.00 |
| 0.20 | 0.20 | 69.00 | 0.20 | 0.18 | 5.33 |
| 0.30 | 0.30 | 111.33 | 0.30 | 0.28 | 6.67 |
| 0.40 | 0.40 | 164.67 | 0.40 | 0.39 | 9.33 |
| 0.50 | 0.50 | 225.33 | 0.50 | 0.44 | 10.00 |
| 0.60 | 0.60 | 295.00 | 0.60 | 0.56 | 12.33 |
| 0.70 | 0.70 | 374.33 | 0.70 | 0.67 | 14.00 |
| 0.80 | 0.80 | 464.33 | 0.80 | 0.78 | 16.67 |
| 0.90 | 0.90 | 561.33 | 0.90 | 0.89 | 19.00 |
| 1.00 | 1.00 | 664.00 | 1.00 | 1.00 | 22.00 |
| Desperf | | | Ethernet | | |
| Percent | Recall | And Reduction (AR) | Percent | Recall | And Reduction (AR) |
| 0.10 | 0.10 | 114.67 | 0.10 | 0.11 | 6.00 |
| 0.20 | 0.21 | 210.33 | 0.20 | 0.18 | 8.00 |
| 0.30 | 0.31 | 318.33 | 0.30 | 0.29 | 10.00 |
| 0.40 | 0.41 | 421.33 | 0.40 | 0.38 | 13.00 |
| 0.50 | 0.50 | 529.67 | 0.50 | 0.47 | 15.00 |
| 0.60 | 0.60 | 657.67 | 0.60 | 0.56 | 16.33 |
| 0.70 | 0.70 | 790.00 | 0.70 | 0.67 | 21.00 |
| 0.80 | 0.80 | 904.67 | 0.80 | 0.79 | 26.67 |
| 0.90 | 0.90 | 1001.33 | 0.90 | 0.89 | 29.67 |
| 1.00 | 1.00 | 1118.00 | 1.00 | 1.00 | 38.00 |
| Conmax | | | Square | | |
| Percent | Recall | And Reduction (AR) | Percent | Recall | And Reduction (AR) |
| 0.10 | 0.10 | 95.00 | 0.10 | 0.11 | 0.00 |
| 0.20 | 0.20 | 188.00 | 0.20 | 0.19 | 0.00 |
| 0.30 | 0.30 | 251.00 | 0.30 | 0.28 | 0.33 |
| 0.40 | 0.40 | 330.67 | 0.40 | 0.38 | 1.33 |
| 0.50 | 0.50 | 411.67 | 0.50 | 0.48 | 2.33 |
| 0.60 | 0.59 | 493.33 | 0.60 | 0.56 | 3.00 |
| 0.70 | 0.69 | 557.67 | 0.70 | 0.65 | 3.67 |
| 0.80 | 0.78 | 625.00 | 0.80 | 0.75 | 4.33 |
| 0.90 | 0.90 | 718.67 | 0.90 | 0.89 | 6.67 |
| 1.00 | 1.00 | 782.00 | 1.00 | 1.00 | 8.00 |

Table 6: We compare CMO with a well-designed GNN COG and the human-designed approach Effisyn. Specifically, we set the hyperparameter $k$ as 50% for COG and our CMO and 70% for Effisyn. And Reduction (AR) denotes the reduced number of nodes, i.e., optimization performance. Normalized AR denotes the ratio of the AR to that of the default heuristic.

| Hyp | | | | Multiplier | | | |
|---|---|---|---|---|---|---|---|
| Method | And Reduction (AR) ↑ | Normalized AR ↑ | Time (s) ↓ | Method | And Reduction (AR) ↑ | Normalized AR ↑ | Time (s) ↓ |
| COG | 661.33 | 1.00 | 247.26 | COG | 21.00 | 0.95 | 18.14 |
| Effisyn | 662.00 | 1.00 | 270.50 | Effisyn | 18.00 | 0.82 | 16.17 |
| CMO (Ours) | 661.00 | 1.00 | **213.39** | CMO (Ours) | 20.00 | 0.91 | **14.32** |
| Square | | | | Desperf | | | |
| Method | And Reduction (AR) ↑ | Normalized AR ↑ | Time (s) ↓ | Method | And Reduction (AR) ↑ | Normalized AR ↑ | Time (s) ↓ |
| COG | 8.00 | 1.00 | 16.11 | COG | 890.67 | 0.80 | 29.97 |
| Effisyn | 1.00 | 0.13 | 16.73 | Effisyn | 895.00 | 0.80 | 26.59 |
| CMO (Ours) | 7.33 | 0.92 | **12.54** | CMO (Ours) | 983.00 | 0.95 | **22.38** |
| Ethernet | | | | Conmax | | | |
| Method | And Reduction (AR) ↑ | Normalized AR ↑ | Time (s) ↓ | Method | And Reduction (AR) ↑ | Normalized AR ↑ | Time (s) ↓ |
| COG | 27.33 | 0.72 | 20.01 | COG | 730.67 | 0.93 | 19.03 |
| Effisyn | 27.00 | 0.71 | 32.85 | Effisyn | 704.00 | 0.90 | 25.29 |
| CMO (Ours) | 30.67 | 0.82 | **16.25** | CMO (Ours) | 703.33 | 0.90 | **15.27** |
| Ci1 | | | | Ci2 | | | |
| Method | And Reduction (AR) ↑ | Normalized AR ↑ | Time (s) ↓ | Method | And Reduction (AR) ↑ | Normalized AR ↑ | Time (s) ↓ |
| COG | 9.67 | 0.81 | 327.05 | COG | 15.50 | 0.67 | 61.89 |
| Effisyn | 9.00 | 0.75 | 275.58 | Effisyn | 19.00 | 0.83 | 59.14 |
| CMO (Ours) | 12.00 | 1.00 | **255.90** | CMO (Ours) | 23.00 | 1.00 | **45.61** |
| Ci3 | | | | Ci4 | | | |
| Method | And Reduction (AR) ↑ | Normalized AR ↑ | Time (s) ↓ | Method | And Reduction (AR) ↑ | Normalized AR ↑ | Time (s) ↓ |
| COG | 1040.00 | 1.00 | 145.12 | COG | 98.00 | 0.99 | 161.19 |
| Effisyn | 1040.00 | 1.00 | 126.31 | Effisyn | 99.00 | 1.00 | **84.29** |
| CMO (Ours) | 1040.00 | 1.00 | **113.08** | CMO (Ours) | 96.00 | 0.97 | 109.42 |
| Sixteen | | | | Twenty | | | |
| Method | And Reduction (AR) ↑ | Normalized AR ↑ | Time (s) ↓ | Method | And Reduction (AR) ↑ | Normalized AR ↑ | Time (s) ↓ |
| COG | 1031.00 | 0.94 | 61905.96 | COG | 1291.00 | 0.95 | 86112.80 |
| Effisyn | 9.00 | 0.01 | 47078.07 | Effisyn | 9.00 | 0.01 | 73853.64 |
| CMO (Ours) | 1000.00 | 0.91 | **32001.27** | CMO (Ours) | 1251.00 | 0.92 | **56965.94** |

Table 7: We present more ablations study results on open-source benchmarks.

| Circuit | Hyp | Multiplier | Square | DesPerf | Ethernet | Conmax |
|---|---|---|---|---|---|---|
| Method | Recall ↑ | Recall ↑ | Recall ↑ | Recall ↑ | Recall ↑ | Recall ↑ |
| CMO | **0.96** | **0.98** | **0.99** | **0.80** | **0.72** | **0.84** |
| CMO without GSD | 0.91 | 0.96 | 0.67 | **0.80** | 0.44 | 0.76 |
| CMO without SFD | **0.99** | 0.80 | 0.80 | 0.73 | 0.66 | 0.42 |
| CMO without SFD and GSD | 0.52 | 0.72 | 0.93 | 0.60 | 0.42 | 0.45 |

Table 8: We compare our CMO with five lightweight baselines. The results demonstrate that our approach outperforms all of the baselines in terms of generalization capability.

| | Hyp | Multiplier | Square | Desperf | Ethernet | Conamx |
|---|---|---|---|---|---|---|
| Method | Recall ↑ | Recall ↑ | Recall ↑ | Recall ↑ | Recall ↑ | Recall ↑ |
| SPL | 0.93 | 0.52 | 0.72 | 0.60 | 0.42 | 0.45 |
| DSR | 0.20 | 0.11 | 0.46 | 0.76 | 0.72 | 0.88 |
| XGBoost | 0.91 | 0.86 | 0.46 | 0.79 | 0.33 | 0.68 |
| RidgeLR | 0.81 | 0.62 | 0.88 | 0.79 | 0.33 | 0.54 |
| Random | 0.50 | 0.44 | 0.48 | 0.50 | 0.47 | 0.50 |
| CMO (Ours) | **0.99** | **0.96** | **0.98** | **0.80** | **0.72** | **0.84** |
| | Sixteen | Twenty | Ci1 | Ci2 | Ci3 | Ci4 |
| Method | Recall ↑ | Recall ↑ | Recall ↑ | Recall ↑ | Recall ↑ | Recall ↑ |
| SPL | 0.43 | 0.44 | 0.85 | 0.91 | 0.98 | 0.80 |
| DSR | 0.63 | 0.63 | 0.85 | 0.83 | 0.95 | 0.94 |
| XGBoost | 0.62 | 0.58 | 0.78 | 0.81 | 0.77 | 0.79 |
| RidgeLR | 0.79 | 0.84 | 0.85 | 0.95 | 0.71 | 0.37 |
| Random | 0.50 | 0.50 | 0.51 | 0.50 | 0.50 | 0.49 |
| CMO (Ours) | **0.86** | **0.85** | **1.00** | **1.00** | **0.99** | **0.96** |

Table 9: We compare our CMO with five lightweight baselines. The results demonstrate that our approach outperforms all of the lightweight baselines in terms of online heuristics efficiency and optimization performance.

| Hyp | | | | Multiplier | | | |
|---|---|---|---|---|---|---|---|
| Method | And Reduction (AR) ↑ | Normalized AR ↑ | Time (s) ↓ | Method | And Reduction (AR) ↑ | Normalized AR ↑ | Time (s) ↓ |
| SPL | 659.33 | 0.99 | 234.88 | SPL | 20.67 | 0.91 | 15.63 |
| DSR | 527.67 | 0.79 | 257.61 | DSR | 4.00 | 0.18 | 14.41 |
| XGBoost | 650.00 | 0.98 | 246.79 | XGBoost | 20.00 | 0.91 | 14.28 |
| RidgeLR | 646.00 | 0.97 | 228.22 | RidgeLR | 20.00 | 0.91 | 11.52 |
| Random | 374.33 | 0.57 | 228.51 | Random | 14.00 | 0.64 | 13.74 |
| GESD (Ours) | 661.00 | **1.00** | **213.39** | GESD (Ours) | 20.00 | **0.91** | **14.32** |
| Square | | | | Desperf | | | |
| Method | And Reduction (AR) ↑ | Normalized AR ↑ | Time (s) ↓ | Method | And Reduction (AR) ↑ | Normalized AR ↑ | Time (s) ↓ |
| SPL | 5.33 | 0.67 | 14.17 | SPL | 927.67 | 0.83 | 31.27 |
| DSR | 1.00 | 0.13 | 20.63 | DSR | 865.00 | 0.77 | 26.42 |
| XGBoost | 1.00 | 0.13 | 19.73 | XGBoost | 1026.00 | 0.92 | **29.97** |
| RidgeLR | 3.00 | 0.38 | 14.90 | RidgeLR | 942.00 | 0.84 | 33.26 |
| Random | 3.67 | 0.46 | 17.82 | Random | 790.00 | 0.71 | 29.42 |
| GESD (Ours) | 7.33 | **0.92** | **12.54** | GESD (Ours) | 983.00 | **0.95** | **22.38** |
| Ethernet | | | | Conmax | | | |
| Method | And Reduction (AR) ↑ | Normalized AR ↑ | Time (s) ↓ | Method | And Reduction (AR) ↑ | Normalized AR ↑ | Time (s) ↓ |
| SPL | 17.67 | 0.46 | 32.39 | SPL | 681.67 | 0.87 | 25.25 |
| DSR | 31.00 | 0.82 | 28.49 | DSR | 767.00 | 0.98 | 22.70 |
| XGBoost | 30.00 | 0.79 | 34.57 | XGBoost | 751.00 | 0.96 | 23.28 |
| RidgeLR | 18.00 | 0.47 | 34.49 | RidgeLR | 638.00 | 0.82 | 24.96 |
| Random | 21.00 | 0.55 | 28.96 | Random | 557.67 | 0.71 | 22.31 |
| GESD (Ours) | 30.67 | **0.82** | **16.25** | GESD (Ours) | 703.33 | **0.90** | **15.27** |

Table 10: we report more results about the Model Inference Time. Note that all of the experiments are tested on purely CPU-based machines.

| | Hyp | Multiplier | Square | Desperf | Ethernet | Conamx |
|---|---|---|---|---|---|---|
| Method | Time (s) ↓ | Time (s) ↓ | Time (s) ↓ | Time (s) ↓ | Time (s) ↓ | Time (s) ↓ |
| COG | 16.730 | 3.415 | 2.383 | 6.238 | 2.771 | 3.557 |
| Effisyn | 0.152 | 0.015 | 0.010 | 0.056 | 0.018 | 0.023 |
| CMO (Ours) | **0.059** | **0.009** | **0.007** | **0.049** | **0.018** | **0.010** |
| | Sixteen | Twenty | Ci1 | Ci2 | Ci3 | Ci4 |
| Method | Time (s) ↓ | Time (s) ↓ | Time (s) ↓ | Time (s) ↓ | Time (s) ↓ | Time (s) ↓ |
| COG | 1377.657 | 1787.070 | 31.710 | 18.230 | 30.540 | 38.770 |
| Effisyn | 9.670 | 12.189 | 0.320 | 0.220 | 0.290 | 0.160 |
| CMO (Ours) | **4.156** | **4.956** | **0.170** | **0.060** | **0.140** | **0.160** |

Table 11: We present the inference times for the GNN model COG and its integrated LO heuristic, COG-Mfs2, on open-source and industrial circuits. Ratio denotes the proportion of the COG model's time accounted for by COG-Mfs2. The results demonstrate that the runtime of the GNN accounts up to 29.46% to that of COG-Mfs2, which significantly impacts the efficiency of the Mfs2 heuristic in purely CPU-based LO tools.

| Hyp | | | Multiplier | | |
|---|---|---|---|---|---|
| COG(time, s) | COG-Mfs2(time, s) | ratio(%) | COG(time, s) | COG-Mfs2(time, s) | ratio(%) |
| 16.73 | 247.26 | **6.77** | 18.14 | 3.42 | **18.83** |
| Square | | | Desperf | | |
| COG(time, s) | COG-Mfs2(time, s) | ratio(%) | COG(time, s) | COG-Mfs2(time, s) | ratio(%) |
| 2.38 | 16.11 | **14.79** | 6.24 | 29.97 | **20.82** |
| Ethernet | | | Conmax | | |
| COG(time, s) | COG-Mfs2(time, s) | ratio(%) | COG(time, s) | COG-Mfs2(time, s) | ratio(%) |
| 2.77 | 20.01 | **13.85** | 3.56 | 19.03 | **18.69** |
| Ci1 | | | Ci2 | | |
| COG(time, s) | COG-Mfs2(time, s) | ratio(%) | COG(time, s) | COG-Mfs2(time, s) | ratio(%) |
| 31.71 | 327.05 | **9.70** | 18.23 | 61.89 | **29.46** |
| Ci3 | | | Ci4 | | |
| COG(time, s) | COG-Mfs2(time, s) | ratio(%) | COG(time, s) | COG-Mfs2(time, s) | ratio(%) |
| 30.54 | 145.12 | **21.04** | 38.77 | 161.19 | **24.05** |

Table 12: A detailed description of circuits from the EPFL benchmark. Nodes denotes the sizes of circuits, and Lev denotes the depths of circuits.

| Circuit | PI | PO | Latch | Nodes | Edge | Cube | Lev |
|---|---|---|---|---|---|---|---|
| Adder | 256 | 129 | 0 | 1020 | 2040 | 1020 | 255 |
| Barrel shifter | 135 | 128 | 0 | 3336 | 6672 | 3336 | 12 |
| Divisor | 128 | 128 | 0 | 57247 | 114494 | 57247 | 4372 |
| Hypotenuse | 256 | 128 | 0 | 214335 | 428670 | 214335 | 24801 |
| Log2 | 32 | 32 | 0 | 32060 | 64120 | 323060 | 444 |
| Max | 512 | 130 | 0 | 2865 | 5730 | 2865 | 287 |
| Multiplier | 128 | 128 | 0 | 27062 | 54124 | 27062 | 274 |
| Sin | 24 | 25 | 0 | 5416 | 10832 | 5416 | 225 |
| Square-root | 128 | 64 | 0 | 24618 | 49236 | 24618 | 5058 |
| Square | 64 | 128 | 0 | 18486 | 36969 | 18485 | 250 |
| Round-robin ariter | 256 | 129 | 0 | 11839 | 23678 | 11839 | 87 |
| Alu control unit | 7 | 26 | 0 | 175 | 348 | 174 | 10 |
| Coding-cavlc | 10 | 11 | 0 | 693 | 1386 | 693 | 16 |
| Decoder | 8 | 256 | 0 | 304 | 608 | 304 | 3 |
| i2c controller | 147 | 142 | 0 | 1357 | 2698 | 1356 | 20 |
| Int to float converter | 11 | 7 | 0 | 260 | 520 | 260 | 16 |
| Memory controller | 1204 | 1230 | 0 | 47110 | 93945 | 47109 | 114 |
| Priority encoder | 128 | 8 | 0 | 978 | 1956 | 978 | 250 |
| Lookahead XY router | 60 | 30 | 0 | 284 | 514 | 257 | 54 |
| Voter | 1001 | 1 | 0 | 13758 | 27516 | 13758 | 70 |

Table 13: A detailed description of circuits from the IWLS benchmark. Nodes denotes the sizes of circuits, and Lev denotes the depths of circuits.

| Circuit | PI | PO | latch | nodes | edge | cube | lev |
|---|---|---|---|---|---|---|---|
| aes_core | 259 | 129 | 530 | 20797 | 40645 | 24444 | 28 |
| des_area | 240 | 64 | 128 | 5005 | 9882 | 5889 | 35 |
| des_perf | 234 | 64 | 8808 | 98463 | 180542 | 108666 | 28 |
| ethernet | 98 | 115 | 10544 | 46804 | 113378 | 72850 | 37 |
| i2c | 19 | 14 | 128 | 1147 | 2299 | 1375 | 15 |
| mem_ctrl | 115 | 152 | 1083 | 11508 | 26436 | 14603 | 31 |
| pci_bridge32 | 162 | 207 | 3359 | 16897 | 34607 | 23130 | 29 |
| pci_conf_cyc_addr_dec | 32 | 32 | 0 | 109 | 212 | 128 | 6 |
| pci_spoci_ctrl | 25 | 13 | 60 | 1271 | 2637 | 1557 | 19 |
| sasc | 16 | 12 | 117 | 552 | 1148 | 766 | 10 |
| simple_spi | 16 | 12 | 132 | 823 | 1694 | 1089 | 14 |
| spi | 47 | 45 | 229 | 3230 | 6904 | 4054 | 32 |
| steppermotordrive | 4 | 4 | 25 | 228 | 397 | 253 | 11 |
| systemcaes | 260 | 129 | 670 | 7961 | 18236 | 11648 | 44 |
| systemcdes | 132 | 65 | 190 | 3324 | 6304 | 3791 | 33 |
| tv80 | 14 | 32 | 359 | 7166 | 16280 | 9352 | 50 |
| usb_funct | 128 | 121 | 1746 | 12871 | 27102 | 16378 | 25 |
| usb_phy | 15 | 18 | 98 | 559 | 1001 | 638 | 12 |
| vga_lcd | 89 | 109 | 17079 | 124050 | 242332 | 146201 | 25 |
| wb_conmax | 1130 | 1416 | 770 | 29036 | 77185 | 39619 | 26 |
| wb_dma | 217 | 215 | 263 | 3495 | 7052 | 4496 | 26 |

Table 14: A detailed description of two very large-scale circuits from the EPFL benchmark. Nodes denotes the sizes of circuits, and Lev denotes the depths of circuits.

| Circuit | PI | PO | Latch | Nodes | Lev |
|---------|-----|-----|-------|----------|-----|
| twenty  | 137 | 60  | 0     | 20732893 | 162 |
| sixteen | 117 | 50  | 0     | 16216836 | 140 |

Table 15: A statsical description of 27 industrial circuits (23 training circuits and 4 testing circuits) from Huawei HiSilicon. Nodes denotes the sizes of circuits, and Lev denotes the depths of circuits.

| Traning Circuits | PI | PO | Latch | Nodes | Lev |
|------------------|----------|----------|-------|----------|----------|
| mean | 8410.5 | 5978.682 | 0 | 104229.4 | 55.95455 |
| max | 59974 | 29721 | 0 | 788288 | 104 |
| min | 41 | 107 | 0 | 2775 | 18 |
| **Testing Circuits** | **PI** | **PO** | **Latch** | **nodes** | **lev** |
| mean | 18540.67 | 18015 | 0 | 356111.2 | 103.3333 |
| max | 42257 | 33849 | 0 | 655243 | 185 |
| min | 523 | 483 | 0 | 24778 | 40 |

Table 16: We present the structural and semantic scoring functions of the industiral circuits to illustrate the interpretability of CMO. Here $x$ represents node feature.

| Ci1 | structural | $\sin(x_0 * x_2 + x_0 - \cos x_1)/x_0$ |
|-----|------------|------------------------------------------|
|     | semantic   | $\neg(x_{64} \wedge x_{20} \wedge x_{24}) \wedge \neg(x_6 \wedge x_{63} \wedge x_{26} \wedge x_{47} \wedge x_{27})$ |
| Ci2 | structural | $\exp x_3 - \cos(x_3/x_0) - x_3 + x_2$ |
|     | semantic   | $\neg(x_{32} \wedge x_{56} \wedge x_{36}) \wedge \neg(x_{47} \wedge \neg x_{33} \wedge x_{57})$ |
| Ci3 | structural | $\sin(x_0 - \cos(x_2) - x_3)/x_0$ |
|     | semantic   | $(x_{52} \vee \neg x_{65}) \wedge \neg(x_{17} \wedge x_{44})$ |
| Ci4 | structural | $x_0 - \exp(x_1)/\exp(x_2)/\sin(x_0 + x_3) + x_3$ |
|     | semantic   | $\neg(x_{26} \wedge x_{62} \wedge x_5 \wedge x_{55}) \wedge \neg(x_{20} \wedge x_{32} \wedge x_{26})$ |

Table 17: We provide our manually designed node features.

| heuristic | Node features | |
|---|---|---|
| Mfs2 | Semantic information | Structure information |
| Input: 6-LUTs | Truth table of the node the node is 6-input LUT (64-dimensional vector) | Fanin number Fanout number Level LevelR Node ID |

Table 18: We compare the Default Mfs2 heuristic with our 2CMO-Mfs2 heuristic with the hyperparameter $k$ set as 30%, 40% and 50% on open-source and industrial circuits. Optimized Nd denotes the node number (size) of circuits, and Lev denotes the level (depth) of circuits. We define an Improvement metric by $\frac{M(\text{Default}) - M(\text{Ours})}{M(\text{Default})}$, where $M(\cdot)$ denotes the Nd, Lev, or Time.

**Open-source Circuits — Hyp / Multiplier**

| Method | Lev ↓ | Improvement ↑ (Optimized Nd, %) | Time (s) ↓ | Improvement ↑ (Time, %) | Optimized Nd ↓ | Improvement ↑ (Optimized Nd, %) | Time (s) ↓ | Improvement ↑ (Time, %) |
|---|---|---|---|---|---|---|---|---|
| Default (Mfs2) | 8259.00 | NA | 319.33 | NA | 7799.00 | NA | 22.58 | NA |
| CMO-Mfs (0.5, Ours) | 8259.00 | 0 | 158.49 | 50.37 | 7801.00 | -0.03 | 14.28 | 36.77 |
| 2CMO-Mfs (0.3, Ours) | 5762.00 | 30.23 | 127.51 | 60.07 | 7661.00 | 7.27 | 18.06 | 20.00 |
| 2CMO-Mfs (0.4, Ours) | 5762.00 | 30.23 | 170.45 | 46.62 | 7659.00 | 7.36 | 20.81 | 7.85 |

**Open-source Circuits — Square / Ethernet**

| Method | Optimized Nd ↓ | Improvement ↑ (Lev, %) | Time (s) ↓ | Improvement ↑ (Time, %) | Optimized Nd ↓ | Improvement ↑ (Optimized Nd, %) | Time (s) ↓ | Improvement ↑ (Time, %) |
|---|---|---|---|---|---|---|---|---|
| Default (Mfs2) | 5701.00 | NA | 27.55 | NA | 13638.00 | NA | 34.48 | NA |
| CMO-Mfs (0.5, Ours) | 5701.67 | -0.01 | 12.54 | 54.47 | 13645.00 | -0.08 | 21.73 | 36.99 |
| 2CMO-Mfs (0.3, Ours) | 5518.67 | 3.26 | 15.04 | 45.42 | 13516.67 | 4.19 | 15.57 | 54.84 |
| 2CMO-Mfs (0.4, Ours) | 5515.33 | 3.32 | 19.06 | 30.80 | 13516.33 | 4.20 | 24.38 | 29.28 |

**Open-source Circuits — Conmax / Desperf**

| Method | Optimized Nd ↓ | Improvement ↑ (Optimized Nd, %) | Time (s) ↓ | Improvement ↑ (Time, %) | Optimized Nd ↓ | Improvement ↑ (Optimized Nd, %) | Time (s) ↓ | Improvement ↑ (Time, %) |
|---|---|---|---|---|---|---|---|---|
| Default (Mfs2) | 16509 | NA | 23.78 | NA | 30853.00 | NA | 36.76 | NA |
| CMO-Mfs (0.5, Ours) | 16587.67 | -0.48 | 15.27 | 35.81 | 30910 | -0.18 | 26.40 | 28.19 |
| 2CMO-Mfs (0.3, Ours) | 15723.33 | 2.00 | 18.24 | 23.32 | 29392.00 | 2.31 | 30.16 | 17.96 |
| 2CMO-Mfs (0.4, Ours) | 15666.67 | 2.08 | 21.65 | 8.95 | 29175.67 | 2.50 | 38.20 | -3.92 |

**very Large-scale Circuits — Sixteen / Twenty**

| Method | Optimized Nd ↓ | Improvement ↑ (Optimized Nd, %) | Time (s) ↓ | Improvement ↑ (Time, %) | Optimized Nd ↓ | Improvement ↑ (Optimized Nd, %) | Time (s) ↓ | Improvement ↑ (Time, %) |
|---|---|---|---|---|---|---|---|---|
| Default (Mfs2) | 6017631 | NA | 78784.04 | NA | 7693089 | NA | 108735.49 | NA |
| CMO-Mfs2 (0.5, Ours) | 6018729 | -0.001 | 32001.27 | 59.38 | 7694455 | -0.002 | 56965.94 | 47.61 |
| 2CMO-Mfs2 (0.3, Ours) | 5434092 | 9.70 | 36425.15 | 53.77 | 6877483 | 10.60 | 59786.55 | 45.02 |
| 2CMO-Mfs2 (0.4, Ours) | 5433745 | 9.70 | 46572.91 | 40.80 | 6877158 | 10.61 | 75956.18 | 30.15 |

**Industrial Circuits — Ci1 / Ci2**

| Method | Lev ↓ | Improvement ↑ (Lev, %) | Time (s) ↓ | Improvement ↑ (Time, %) | Optimized Nd ↓ | Improvement ↑ (Optimized Nd, %) | Time (s) ↓ | Improvement ↑ (Time, %) |
|---|---|---|---|---|---|---|---|---|
| Default (Mfs2) | 47 | NA | 180.35 | NA | 99245 | NA | 78.07 | NA |
| CMO-Mfs2 (0.5, Ours) | 47 | 0.00 | 113.08 | 37.30 | 99245 | 0.00 | 45.61 | 41.58 |
| 2CMO-Mfs2 (0.3, Ours) | 45 | 4.26 | 142.96 | 20.73 | 94240 | 5.04 | 64.37 | 17.55 |
| 2CMO-Mfs2 (0.4, Ours) | 45 | 4.26 | 177.04 | 1.84 | 93184 | 6.11 | 81.07 | -3.84 |

**Industrial Circuits — Ci3 / Ci4**

| Method | Optimized Nd ↓ | Improvement ↑ (Optimized Nd, %) | Time (s) ↓ | Improvement ↑ (Time, %) | Optimized Nd ↓ | Improvement ↑ (Optimized Nd, %) | Time (s) ↓ | Improvement ↑ (Time, %) |
|---|---|---|---|---|---|---|---|---|
| Default (Mfs2) | 196565 | NA | 458.31 | NA | 215708 | NA | 201.43 | NA |
| CMO-Mfs2 (0.5, Ours) | 196565 | 0.00 | 255.90 | 44.16 | 215711 | -0.001 | 109.42 | 45.68 |
| 2CMO-Mfs2 (0.3, Ours) | 194516 | 1.042 | 402.67 | 12.140 | 215476 | 0.11 | 182.60 | 9.35 |
| 2CMO-Mfs2 (0.4, Ours) | 194516 | 1.042 | 515.15 | -12.402 | 215476 | 0.11 | 202.72 | -0.64 |

