# OpenReview forum: "A Graph Enhanced Symbolic Discovery Framework For Efficient Logic Optimization"
_ICLR.cc/2025/Conference — ICLR 2025 Poster_

### Official Review · Reviewer_kVA7 · 2024-10-27

**Soundness:** 3
**Presentation:** 2
**Contribution:** 2
**Rating:** 8
**Confidence:** 4

**Summary:**

This paper proposes a method called CMO to develop lightweight and generalizable scoring functions for ranking nodes in an AIG, aiming to enhance the efficiency and performance of logic optimization. The method is clearly introduced. The paper trains a GNN and uses an MCTS-based symbolic regression method to generate symbolic scoring functions, ensuring both inference efficiency and generalization. However, some experimental details remain unclear.

**Strengths:**

The method is clearly introduced, with comprehensive experiments demonstrating the effectiveness and performance of CMO.

**Weaknesses:**

### Topic
The term "circuit synthesis" is not well-defined in the field of EDA, which may cause confusion. Based on the related works and experiments, this paper appears to focus on logic optimization.
### Datasets
The labels of circuit datasets should be clarified. When mentioning node-level transformation, does it mean it is effective for the current step of logic optimization or for overall performance? Effectiveness in the current step may not translate to overall performance in logic optimization.
### Experiments
The experiment part is somewhat confusing. The focus of logic optimization should be on time cost and node reduction during the online phase. The offline phase appears more like an ablation study. Experiments on generalization should be highlighted in the main part of the manuscript. Experiment 4 should showcase generalization compared to other baselines like COG, but why other SR methods? Is this an ablation study of the SR method used?
### Generalization
EPFL, IWLS, and an industrial-level dataset from Huawei HiSilicon are used to train the GNN. Are the datasets mixed to train a single GNN, or are three separate GNNs trained for each dataset?

**Questions:**

1. Can CMO generalize to other logic optimization methods like Rewrite?
2. Can a GNN trained on one dataset generalize to another dataset?
3. How are the training dataset labels obtained?
4. The presentation of experiments is confusing. Please clarify which is the main experiment and which are ablation studies.

---

> ### Author Response · Authors · 2024-11-22
> **Rebuttal by Authors**
>
> # Response to Reviewer kVA7
> We thank the reviewer for the insightful and valuable comments. We respond to each comment as follows and sincerely hope that our rebuttal can properly address your concerns. If so, we would deeply appreciate it if you could raise your score. If not, please let us know your further concerns, and we will continue actively responding to your comments and improving our submission.
>
> ### Weakness 1
> > **The term "circuit synthesis" is unclear in EDA and may cause confusion; this paper seems to focus on logic optimization based on related works and experiments.**
>
> Thanks for your valuable comments. In our initial draft, **we followed previous works [1][2][3][4]** in adopting the term "Circuit Synthesis" as a more accessible alternative to "Logic Synthesis" to enhance readability for non-experts in the field." In general, Logic Synthesis consists of three main stages: translation, logic optimization, and technology mapping. As you insightfully mentioned, our work mainly focuses on logic optimization and it is more proper to use logic optimization rather than circuit synthesis. Therefore, to ensure accuracy and clarity, **we have replaced "Circuit Synthesis" with "Logic Optimization" in the first revision.**
>
> [1]. Wang Z et al. Towards Next-Generation Logic Synthesis: A Scalable Neural Circuit Generation Framework. The Thirty-eighth Annual Conference on Neural Information Processing Systems. 2024.
>
> [2]. Chowdhury, et al. Openabc-d: A large-scale dataset for machine learning guided integrated circuit synthesis. arXiv preprint arXiv:2110.11292, 2021.
>
> [3]. Scarabottolo I, Ansaloni G, Constantinides G A, et al. Approximate logic synthesis: A survey. Proceedings of the IEEE, 2020, 108(12): 2195-2213.
>
> [4]. Buch, et al. Logic synthesis for large pass transistor circuits. 1997 Proceedings of IEEE International Conference on Computer Aided Design (ICCAD). IEEE, 1997: 663-670.
>
> ### Weakness 2 & Question 3
> > **How are the training dataset labels obtained? When mentioning node-level transformation, does it mean it is effective for the current step of logic optimization or for overall performance? Effectiveness in the current step may not translate to overall performance in logic optimization**
>
> The node label $y $ is collected based on the effectiveness of the node-level transformation. **If the node-level transformation is effective at the node $\textbf{x} $, then $y=1 $. Otherwise, $y=0 $.**
>
> The effectiveness of a node-level transformation is determined by whether the heuristic **can optimize the current local subgraph (i.e., reducing the subgraphs nodes)**. While some locally effective nodes may influence overall optimization performance, others may not. However, the proportion of **locally effective nodes that fail to contribute to overall performance is sufficiently small**, so labeling these nodes as positive **has no significant impact efficiency**. Moreover, applying node-level transformations on these nodes **will not degrade the final optimization performance**.  Therefore, it is reasonable and simple to label nodes based on the local effectiveness of the node-level transformations.
>
> ### Weakness 3.1
> > **The focus of logic optimization should be on time cost and node reduction during the online phase. The offline phase appears more like an ablation study.**
>
> Thanks for your valuable comments. Our CMO framework consists of two phases: the offline phase and the online phase. In the offline phase, we formulate the logic optimization (LO) task **as a machine learning (ML) problem**---learning a model from the training dataset that can accurately predict ineffective nodes on unseen circuits. **The offline metric**---prediction recall, which serves as a proxy for the online optimization performance---**is used to access the generalization capability of the learned models from the ML perspective**. In the online phase, the CMO uses the learned model to predict ineffective nodes on unseen circuits and avoid transformations on these nodes to accelerate the X heuristic. **The online generalization metrics**---runtime and node reduction---**are used to evaluate the efficiency and generalization optimization performance of the heuristic X-Mfs2 from th EDA perspective**.
>
> In Experiment 1, the offline results are used to show the superior generalization performance of our CMO. In terms of online results, **we mainly focus on time cost and node reduction** as you suggested. Specifically, **We present the online results in Figure 4 and Table 6 in the initial submission.** Table 6 is provided in Appendix C.3 due to limited space. For your convenience, we have included Table 6 below. The results show that **our CMO significantly outperforms the baselines in terms of efficiency while achieving comparable optimization performance with the GPU-based SOTA methods COG.**

---

> > ### Author Response · Authors · 2024-11-22
> >
> > **Table 6:** We compare CMO with the GPU-based SOTA approach COG and the human-designed approach Effisyn. Specifically, we set the Top $k $ as $50 $% for the COG and CMO and $70 $% for the Effisyn. And Reduction (AR) denotes the reduced number of nodes (optimization performance). Normalized AR denotes the ratio of the AR to that of the default heuristic.
> > | **Hyp**   |   |   |   | **Multiplier**   |   |   |   |
> > |--------------|--------------------------|---------------------|----------------|--------------|--------------------------|---------------------|----------------|
> > | Method   | And Reduction (AR)  | Normalized AR  | Time (s) | Method   | And Reduction (AR) | Normalized AR  | Time (s)  |
> > | COG      | 661.33                   | 1.00                | 247.26         | COG        | 21.00        | **0.95**                     | 18.14               |
> > | Effisyn  | 662.00                   | 1.00                | 270.50         | Effisyn    | 18.00        | 0.82                     | 16.17               |
> > | CMO (Ours)| 661.00                   | **1.00**                | **213.39**         | CMO (Ours) | 20.00        | 0.91                     | **14.32**               |
> > | **Square**   |   |   |   | **Desperf**   |   |   |   |
> > | Method   | And Reduction (AR)  | Normalized AR  | Time (s) | Method   | And Reduction (AR) | Normalized AR  | Time (s)  |
> > | COG      | 8.00                     | **1.00**                | 16.11          | COG        | 890.67       | 0.80                     | 29.97               |
> > | Effisyn  | 1.00                     | 0.13                | 16.73          | Effisyn    | 895.00       | 0.80                     | 26.59               |
> > | CMO (Ours)| 7.33                     | 0.92                | **12.54**          | CMO (Ours) | 983.00       | **0.95**                     | **22.38**               |
> > | **Ethernet**   |   |   |   | **Conmax**   |   |   |   |
> > | Method   | And Reduction (AR)  | Normalized AR  | Time (s) | Method   | And Reduction (AR) | Normalized AR  | Time (s)  |
> > | COG      | 27.33                    | 0.72                | 20.01          | COG        | 730.67       | **0.93**                     | 19.03               |
> > | Effisyn  | 27.00                    | 0.71                | 32.85          | Effisyn    | 704.00       | 0.90                     | 25.29               |
> > | CMO (Ours)| 30.67                    | **0.82**                | **16.25**          | CMO (Ours) | 703.33       | 0.90                     | **15.27**               |
> > | **Ci1**   |   |   |   | **Ci2**   |   |   |   |
> > | Method   | And Reduction (AR)  | Normalized AR  | Time (s) | Method   | And Reduction (AR) | Normalized AR  | Time (s)  |
> > | COG      | 9.67                     | 0.81                | 327.05         | COG        | 15.50        | 0.67                     | 61.89               |
> > | Effisyn  | 9.00                     | 0.75                | 275.58         | Effisyn    | 19.00        | 0.83                     | 59.14               |
> > | CMO (Ours)| 12.00                    | **1.00**                | **255.90**         | CMO (Ours) | 23.00        | **1.00**                     | **45.61**               |
> > | **Ci3**   |   |   |   | **Ci4**   |   |   |   |
> > | Method   | And Reduction (AR)  | Normalized AR  | Time (s) | Method   | And Reduction (AR) | Normalized AR  | Time (s)  |
> > | COG      | 1040.00                  | 1.00                | 145.12         | COG        | 98.00        | 0.99                     | 161.19              |
> > | Effisyn  | 1040.00                  | 1.00                | 126.31         | Effisyn    | 99.00        | **1.00**                     | 84.29               |
> > | CMO (Ours)| 1040.00                  | **1.00**                | **113.08**         | CMO (Ours) | 96.00        | 0.97                     | **109.42**              |
> > | **Sixteen**   |   |   |   | **Twenty**   |   |   |   |
> > | Method   | And Reduction (AR)  | Normalized AR  | Time (s) | Method   | And Reduction (AR) | Normalized AR  | Time (s)  |
> > | COG      | 1031.00                  | **0.94**                | 61905.96       | COG        | 1291.00      | **0.95**                     | 86112.80            |
> > | Effisyn  | 9.00                     | 0.01                | 47078.07       | Effisyn    | 9.00         | 0.01                     | 73853.64            |
> > | CMO (Ours)| 1000.00                  | 0.91                | **32001.27**       | CMO (Ours) | 1251.00      | 0.92                     | **56965.94**            |

---

> ### Author Response · Authors · 2024-11-22
>
> ### Weakness 3.2
> > **Experiments on generalization should be highlighted in the main part of the manuscript.**
>
> Thanks for your valuable comments. According to your suggestion, we have revised the experiment section **in the first revision (Line 454)** by **removing the generalization results in Experiment 4** and **organizing all the generalization comparison results in Experiment 1** to better emphasize the generalization evaluation in the main part. For your convenience, we summarize the adjusted experiment sections as follows:
> - **Experiment 1. To demonstrate the superior performance of our CMO in terms of generalization performance and efficiency.**
> - Experiment 2. To demonstrate that our approach can not only prompt the efficiency of the Mfs2 heuristic but also improve the Quality of Results (QoR).
> - Experiment 3. Perform carefully designed ablation studies to provide further insight into CMO.
> - **Experiment 4. To show the appealing features of CMO in terms of ~~generalization capability~~ per inference efficiency and interpretability.**
>
> ### Weakness 3.3
> > **Experiment 4 should showcase generalization compared to other baselines like COG, but why other SR methods? Is this an ablation study of the SR method used?**
>
> We sincerely apologize for the confusion caused by our experiment setting. **The five lightweight methods** discussed in Experiment 4 are **baselines for generalization and efficiency evaluation in Experiment 1, rather than being part of the ablation studies**.
>
> Specifically, we have mentioned in 'Evaluation Metrics and Evaluated Methods' that, **in the offline and online phase, we compare our CMO with the other five lightweight baselines  (Line 372 in the initial submission)** to provide a more comprehensive comparison. However, these comparisons were not appropriately placed in the correct section of the Experiment.
>
>
> To clarify this misunderstanding, we have **removed the "High Generalization Performance" section from Experiment 4** and incorporated the comparison results with these lightweight methods into Experiment 1. The updated results for the generalization and online heuristics efficiency comparisons are now presented in **Tables 8 and 9 in the Appendix of the first revision.** For your convenience, we have included the tables below:
>
> **Table 8**: We compare our CMO with five lightweight baselines. The results demonstrate that our approach outperforms all of the baselines in terms of generalization capability.
> |            | Hyp    | Multiplier | Square | Desperf | Ethernet | Conmax |
> |------------|--------|------------|--------|---------|----------|--------|
> | Method     | Recall | Recall     | Recall | Recall  | Recall   | Recall |
> | SPL        | 0.93   | 0.52       | 0.72   | 0.60    | 0.42     | 0.45   |
> | DSR        | 0.20   | 0.11       | 0.46   | 0.76    | 0.72     | 0.88   |
> | XGBoost    | 0.91   | 0.86       | 0.46   | 0.79    | 0.33     | 0.68   |
> | RidgeLR    | 0.81   | 0.62       | 0.88   | 0.79    | 0.33     | 0.54   |
> | Random     | 0.50   | 0.44       | 0.48   | 0.50    | 0.47     | 0.50   |
> | CMO (Ours) | **0.99**   | **0.97**       | **0.98**   | **0.80**    | **0.72**     | **0.84**   |

---

> > ### Author Response · Authors · 2024-11-22
> >
> > **Table 9**: The results demonstrate that our approach outperforms all of the lightweight baselines in terms of online heuristics efficiency and optimization performance.
> >
> > | Hyp        |                    |               |             | Multiplier |                    |               |              |
> > |------------|--------------------|---------------|-------------|------------|--------------------|---------------|--------------|
> > | Method     | And Reduction (AR) | Normalized AR | Time (s)    | Method     | And Reduction (AR) | Normalized AR | Time (s)     |
> > | SPL        | 659.33             | 0.99          | 234.88      | SPL        | 20.67              | 0.91          | 15.63        |
> > | DSR        | 527.67             | 0.79          | 257.61      | DSR        | 4.00               | 0.18          | 14.41        |
> > | XGBoost    | 650.00             | 0.98          | 246.79      | xgboost    | 20.00              | 0.91          | 14.28        |
> > | RidgeLR    | 646.00             | 0.97          | 228.22      | RidgeLR    | 20.00              | 0.91          | 11.52        |
> > | Random     | 374.33             | 0.57          | 228.51      | Random     | 14.00              | 0.64          | 13.74        |
> > | CMO (Ours) | 661.00             | **1.00**      | **213.39**  | CMO (Ours) | 20.00              | **0.91**          | **14.32**    |
> > | **Square** |                    |               |             | **Desperf** |                    |               |              |
> > | Method     | And Reduction (AR) | Normalized AR | Time (s)    | Method      | And Reduction (AR) | Normalized AR | Time (s)     |
> > | SPL        | 5.33               | 0.67          | 14.17       | SPL         | 927.67             | 0.83          | 31.27        |
> > | DSR        | 1.00               | 0.13          | 20.63       | DSR         | 865.00             | 0.77          | 26.42        |
> > | XGBoost    | 1.00               | 0.13          | 19.73       | xgboost     | 1026.00            | 0.92          | 29.97        |
> > | RidgeLR    | 3.00               | 0.38          | 14.90       | RidgeLR     | 942.00             | 0.84          | 33.26        |
> > | Random     | 3.67               | 0.46          | 17.82       | Random      | 790.00             | 0.71          | 29.42        |
> > | CMO (Ours) | 7.33               | **0.92**          | **12.54**   | CMO (Ours)  | 983.00             | **0.95**      | **22.38**    |
> > | **Ethernet**   |                    |               |             | **Conmax**      |                    |               |              |
> > | Method     | And Reduction (AR) | Normalized AR | Time (s)    | Method      | And Reduction (AR) | Normalized AR | Time (s)     |
> > | SPL        | 17.67              | 0.46          | 32.39       | SPL         | 681.67             | 0.87          | 25.25        |
> > | DSR        | 31.00              | 0.82          | 28.49       | DSR         | 767.00             | 0.98          | 22.70        |
> > | XGBoost    | 30.00              | 0.79          | 34.57       | xgboost     | 751.00             | 0.96          | 23.28        |
> > | RidgeLR    | 18.00              | 0.47          | 34.49       | RidgeLR     | 638.00             | 0.82          | 24.96        |
> > | Random     | 21.00              | 0.55          | 28.96       | Random      | 557.67             | 0.71          | 22.31        |
> > | CMO (Ours) | 30.67              | **0.82**      | **16.25**   | CMO (Ours)  | 703.33             | **0.90**          | **15.27**    |

---

> > > ### Author Response · Authors · 2024-11-22
> > >
> > > ### Weakness 4
> > > > **Are the datasets mixed to train a single GNN, or are three separate GNNs trained for each dataset?**
> > >
> > > In our approach, we use the leave-one-out generalization evaluation strategy and **train separate GNNs for each dataset.** For instance, given the EPFL benchmark, we construct a Log2 dataset by designating Log2 as the testing dataset and using the remaining circuits (including Hyp and Square) in the EPFL benchmark as the training dataset. A GNN model is then trained on the training dataset and employed to discover symbolic functions for the Log2 dataset.
> > >
> > > ### Question 1
> > > > **Can CMO generalize to other logic optimization methods?**
> > >
> > > **Yes, our CMO framework can generalize to other logic optimization (LO) methods.** The common LO heuristics include **Rewrite [5], Refactor [6], Resub [7], and Mfs2 [8]**, which **all follow the paradigm shown in Figure 5 in the initial submission [1]**. Due to time constraints, we only tested our method on **the most time-consuming heuristics, i.e., resub, among all the heuristics (see Table b).** However, we believe that our method can generalize to other heuristics as they follow the same paradigm and are only different in the node-level transformations.
> > >
> > > To verify whether our method can generalize to the resub heuristic, our CMO **first follows [1] to collect training dataset** $\mathcal{D} = \lbrace \textbf{x}\_i, y_i \rbrace_{i=1}^N $. The node features are obtained from an AIG that contains structural and semantic information, and the labels are collected based on the effectiveness of the node-level transformations. **Then we train a GNN model** on the training dataset and employ our GESD framework to distill a symbolc function from the teacher GNN model. **Finally, we evaluate the generalization performance of the learned symbolic functions.** The results in Table c demonstrate that our CMO **achieves an average prediction recall of 85% on the test circuits.** Therefore, we can conclude that our CMO effectively generalizes to other logic optimization heuristics such as Resub.
> > >
> > > **Table b:** We analyze the runtime of commonly used LO heuristics on six challenging open-source circuits. The ratio denotes the ratio of the runtime to that of the Rewrite heuristic.
> > > |Avg Time Ratio to Rewrite             |         |         | |       |       |
> > > |------------|---------|---------|---------------------------|-------|-------|
> > > | Heuristics | Rewrite | Balance | Refactor                  | Resub | Mfs2  |
> > > | Time Ratio | 1       | 0.05    | 1.21                      | **73.44** | 30.94 |
> > >
> > > **Table c:** We evaluate our CMO, COG, and Effisyn on six challenging open-source circuits. The results demonstrate that our CMO can generalize to the pre-mapping heuristic Resub.
> > > |            | Hyp   | Multiplier | Square | Des_perf | Ethernet | Conmax | Average  |
> > > |------------|-------|------------|--------|----------|----------|-----------|----------|
> > > | Method     | Recall | Recall      | Recall  | Recall    | Recall    | Recall     | Recall    |
> > > | COG        | 0.90  | **0.95**       | 0.82   | **0.79**     | **0.99**     | 0.76      | **0.87**     |
> > > | Effisyn    | 0.67  | 0.20       | 0.63   | 0.46     | 0.82     | 0.020     | 0.47     |
> > > | CMO (ours) | **0.90**  | 0.88       | **0.82**   | 0.65     | 0.91     | **0.92**      | **0.85**     |
> > >
> > > [5]. Bertacco, Damiani. The disjunctive decomposition of logic functions. 1997 Proceedings of IEEE International Conference on Computer Aided Design (ICCAD). IEEE, 1997: 78-82.
> > >
> > > [6]. Brayton R K. The decomposition and factorization of Boolean expressions. ISCA-82, 1982: 49-54.
> > >
> > > [7]. Brayton A M R. Scalable logic synthesis using a simple circuit structure. Proc. IWLS. 2006, 6: 15-22.
> > >
> > > [8]. Mishchenko A, et al. Scalable don't-care-based logic optimization and resynthesis. ACM Transactions on Reconfigurable Technology and Systems (TRETS), 2011, 4(4): 1-23.

---

> > > > ### Author Response · Authors · 2024-11-22
> > > >
> > > > ### Question 2
> > > > > **Can a GNN trained on one dataset generalize to another dataset?**
> > > >
> > > > **Yes, the GNN trained on one dataset can generalize to another dataset.** Specifically, we trained a GNN model using **all of the circuits from the IWLS benchmark** and tested it on five challenging circuits **from the EPFL benchmark**. The results in **Table d** show that **the GNN trained on one dataset successfully generalizes to another dataset for both Mfs2 and Resub heuristics.**
> > > >
> > > > **Tabel d:** We trained the models from IWLS benchmarks and generalize them to EPFL circuits. The results demonstrate that the GNN achieves high benchmark-generalization performance for both Mfs2 and Resub heuristics.
> > > > |                 |          |             |              |             |             |              |
> > > > |-----------------|----------|-------------|--------------|-------------|-------------|--------------|
> > > > | **Mfs2 heuristic**  | **Circuits** | **Hyp**         | **Log2**         | **Multiplier**  | **Sin**         | **Square**       |
> > > > |                 | Method   | Recall      | Recall       | Recall      | Recall      | Recall       |
> > > > |                 | COG      | **0.85** | **0.88**  | **0.87** | **0.79** | **0.58**  |
> > > > |                 | Effisyn  | 0.34        | 0.09         | 0.61        | 0.95        | 0.55         |
> > > > |                 | Random   | 0.50        | 0.46         | 0.44        | 0.47        | 0.48         |
> > > > | **Resub heuristic**  | **Circuits** | **Hyp**         | **Log2**         | **Multiplier**  | **Sin**         | **Square**       |
> > > > |                 | Method   | Recall      | Recall       | Recall      | Recall      | Recall       |
> > > > |                 | COG      | **0.87** | **0.80**  | **0.87**  | **0.81** | **0.89**   |
> > > > |                 | Effisyn  | 0.65        | 0.47         | 0.2         | 0.61        | 0.57         |
> > > > |                 | Ramdom   | 0.5         | 0.48         | 0.58        | 0.54        | 0.47         |
> > > >
> > > >
> > > > ### Question 4
> > > > > **The presentation of experiments is confusing. Please clarify which is the main experiment and which are ablation studies.**
> > > >
> > > > Thanks for your valuable comments. **The main experiment is Experiment 1—Generalization and Efficiency Evaluation, while the ablation studies are presented in Experiment 3, comparing CMO with CMO without GESD and CMO without SFD and GESD.** We have clarified in Weakness 3.1, 3.2, and 3.3 how we revised our Experiment Section. Once again, we appreciate your insightful suggestions, which have greatly improved the clarity of our paper.

---

> > > > > ### Author Response · Authors · 2024-11-24
> > > > > **We would love to hear your feedback**
> > > > >
> > > > > Dear Reviewer kVA7,
> > > > >
> > > > > We greatly appreciate your careful reading and constructive comments! We sincerely hope that our rebuttal **has properly addressed all your concerns**, including **revisions to clarify the Experiment section** (see *Weakness 3 and Question 4*), **addtitional generalization results** (see *Question 1 and Question 2*), and **explanation for our label collection strategy** (see *Weakness 2 and Question 3*). Item-by-item responses to your comments are provided above this response for your reference.
> > > > >
> > > > > As the deadline for the author-reviewer discussion period is approaching (due on Novemeber 27), and **we are looking forward to your feedback and/or questions**! We would deeply appreciate it if you could raise your score if our rebuttal has addressed your concerns. If not, please let us know your further concerns, and we will continue actively responding to your comments and improving our submission.
> > > > >
> > > > > Best,
> > > > >
> > > > > Authors

---

> > > > > > ### Author Response · Authors · 2024-11-28
> > > > > > **We eagerly await your feedback**
> > > > > >
> > > > > > Dear Reviewer kVA7,
> > > > > >
> > > > > > We are writing to gently remind you that **the deadline for the author-reviewer discussion period is approaching** (due on December 2nd). We eagerly await your feedback to understand if our responses have adequately addressed all your concerns. *If so, we would deeply appreciate it if you could raise your score*. If not, please let us know your further concerns, and we will continue actively responding to your comments and improving our submission. We sincerely thank you once more for your insightful comments and kind support.
> > > > > >
> > > > > > Best,
> > > > > >
> > > > > > Authors

---

> > > > > > > ### Author Response · Authors · 2024-11-30
> > > > > > > **We would greatly appreciate hearing your feedback**
> > > > > > >
> > > > > > > Dear Reviewer kVA7,
> > > > > > >
> > > > > > > We are writing to kindly remind you that the deadline for the author-reviewer discussion period is fast approaching (**ending in two days** on December 2nd). We greatly value your feedback and are eager to hear your thoughts on whether our responses have sufficiently addressed your concerns. Thank you once again for your thoughtful comments and valuable support throughout this process.
> > > > > > >
> > > > > > > Best,
> > > > > > >
> > > > > > > Authors

---

> > > > > > > > ### Author Response · Authors · 2024-12-02
> > > > > > > > **We would greatly appreciate hearing your feedback.**
> > > > > > > >
> > > > > > > > Dear Reviewer kVA7,
> > > > > > > >
> > > > > > > > We would like to express our sincere gratitude once again for your positive feedback, insightful comments, and constructive suggestions. Your guidance has been invaluable in helping us improve the quality of our work!
> > > > > > > >
> > > > > > > > We are writing to gently remind you that **the author-reviewer discussion period will end in less than 36 hours**. We eagerly await your feedback to **understand if our responses have adequately addressed your concerns**. **If so, we would deeply appreciate it if you could raise your score**. If not, we are eager to address any additional queries you might have, which will enable us to further enhance our work.
> > > > > > > >
> > > > > > > > Once again, thank you for your kind support and constructive suggestions!
> > > > > > > >
> > > > > > > > Best,
> > > > > > > >
> > > > > > > > Authors

---

> > > > > > > > > ### Author Response · Authors · 2024-12-03
> > > > > > > > > **We would greatly appreciate hearing your feedback**
> > > > > > > > >
> > > > > > > > > Dear Reviewer kVA7,
> > > > > > > > >
> > > > > > > > > We would like to sincerely thank you once again for your positive feedback, insightful comments, and constructive suggestions. Your guidance has been invaluable in enhancing the quality of our work!
> > > > > > > > >
> > > > > > > > > As the author-reviewer discussion period is approaching its conclusion with **less than 12 hours** remaining, we wanted to kindly follow up regarding your feedback. We are eager to hear your thoughts on **whether our responses have sufficiently addressed your concerns**. **If they have, we would greatly appreciate it if you could consider reflecting this in your score.** If there are any remaining questions or concerns, we would be more than happy to provide further clarifications within the remaining time.
> > > > > > > > >
> > > > > > > > > Thank you again for your support and thoughtful guidance throughout this process. We deeply value your time and effort in reviewing our work.
> > > > > > > > >
> > > > > > > > > Best，
> > > > > > > > >
> > > > > > > > > Authors

---

> > > > > > > > > > ### Comment · Reviewer_kVA7 · 2024-12-03
> > > > > > > > > >
> > > > > > > > > > I appreciate the effort made by the authors!
> > > > > > > > > >
> > > > > > > > > > My main concerns previously were the misused terms, vague details, and the lack of baselines. Most of them have been addressed in the rebuttal or revised in the manuscript by the authors.
> > > > > > > > > >
> > > > > > > > > > Thus, I decided to raise my score to 7/8. I hope the authors can retain these revisions in the manuscript and make them clear.

---

> ### Author Response · Authors · 2024-12-03
> **Appreciation for your feedback**
>
> Dear Reviewer kVA7,
>
> We are truly grateful for your kind support and the time you've taken to review and assess our work. Your wiilingness to consider raising the score means a great deal to us and gives us further confidence in the value of our contributions. According to your suggestions, we will retain the revisions in the manuscript to make them clear.
>
> Best,
>
> Authors

---

> > ### Author Response · Authors · 2024-12-04
> > **Thank you!**
> >
> > Thank you for your kind support and valuable feedback on our paper! We appreciate your insightful comments and constructive suggestions.

---

### Official Review · Reviewer_oLyS · 2024-10-28

**Soundness:** 3
**Presentation:** 3
**Contribution:** 3
**Rating:** 6
**Confidence:** 3

**Summary:**

This paper studies the problem of circuit synthesis (CS) via graph-based methods that can generalize to unseen circuits. The proposed method, CMO, combines symbolic function learning with Graph Neural Networks.

**Strengths:**

1. The paper is well-written with good introduction to the domain especially for ML-audience less familiar with hardware design.
2. Choice of benchmarks and state-the-art heuristics seem to be solid with comprehensive evaluation.
3. The proposed method achieves significant speedup while maintaining optimization performance on real circuits.

**Weaknesses:**

1. Theoretical justification and analysis are lacking – It seems combining GNN, MCTS, symbolic learning etc. leads to better results on these CS benchmarks, yet some deeper explanation and analysis can be provided to make the paper stronger.
2. Some of the writings can be improved, e.g. “However, this approach cannot capture effective information from specific circuit distribution for higher generalization performance” – Is it due to the human-designed nature and lack of adoption of machine learning from existing data?
3. Some technical errors, e.g. “Specifically, we use mean absolute error and focal loss” yet the equation (4) is an MSE loss.
4. More circuit dataset descriptions, e.g. graph sizes, and graph visualizations would provide a more solid background for ML audience.

**Questions:**

N/A

---

> ### Author Response · Authors · 2024-11-22
> **Rebuttal by Authors**
>
> # Response to Reviewer oLyS
> We thank the reviewer for the insightful and valuable comments. We respond to each comment as follows and sincerely hope that our rebuttal can properly address your concerns. If so, we would deeply appreciate it if you could raise your score. If not, please let us know your further concerns, and we will continue actively responding to your comments and improving our submission.
>
> ###  Weakness 1.
> > **Provide some deeper explanation and theoretical analysis on why combining GNN, MCTS, and symbolic learning leads to better results.**
>
> Thanks for your valuable comments. To offer a deeper explanation of why the combination of GNN, MCTS, and symbolic learning leads to improved results, we break the analysis down into the following four steps:
>
> **Step 1. We provide a theoretical analysis of how the teacher GNN model addresses the circuit generalization problem in our CS task.**
>
> The GNN addresses the circuit generalization problem by **designing multi-domain circuit datasets for training** and **learning domain-invariant information (i.e., inductive bias) from well-constructed subgraphs.** Specifically, [1] first discovers that the large distribution shift between different circuits makes it challenging for models trained on training circuits to generalize to the unseen circuits. To address this problem, [1] provides a theorem **for the generalization error bound** between the average risk $\mathcal{R}(f) $ over all possible target circuit domains and the empirical risk estimation objective $\hat{\mathcal{R}}(f) $ on the training circuit domains. A circuit domain refers to the underlying distribution from which circuits are sampled.
>
> **Theorem 1**: Under some mild and reasonable assumptions, the following inequality holds with probability at least 1 - $\delta $
>
> $$\left(\sup\_{f}\lvert \mathcal{R}(f)-\hat{\mathcal{R}}(f) \rvert \right)^2 \leq \vphantom{\frac{C_4}{M^2}\sum_{k=1}^M\frac{1}{n_k}}\frac{C_1 \log \delta^{-1}+C_2}{M} + \frac{C_3\log 2\delta^{-1}M+C_4\log \delta^{-1}+C_5}{M^2}\sum_{k=1}^M\frac{1}{n_k} $$
>
> where $C_1, C_2, C_3, C_4, C_5 $ are constants, $M $ is the number of training circuit domains and $n_k $ is the sample size of the $k $-th training circuit domain. In our CS task, the total number of the training samples $n = \sum_{k=1}^M{n_k} $ is a constant. Based on Theorem 1, we derive the following corollary:
>
> **Corollary 1**: Under the condition $n_k = \frac{n}{M} \text{ for } k = 1, 2, \cdots, M $ and $1 \leq M \leq \frac{C_1\log\delta^{-1}+C_2} {C_3\log2\delta^{-1}} \cdot n $, **using domain-wise training circuit datasets (i.e., M > 1) will result in a smaller generalization error bound than just pooling them in one mixed dataset (i.e., M=1).** The proof is provided below:
>
> We represent the generalization error bound in Theorem 1 as a function on discrete variable $M $
>     \begin{align}
>         B(M) = \frac{C_1 \log \delta^{-1}+C_2}{M}+ \frac{C_3\log 2\delta^{-1}M}{n}+\frac{C_4\log \delta^{-1}+C_5}{n}\quad(M \geq 1)\nonumber
>     \end{align}
> where $n $ representing the total number of samples is a constant and $M $ denotes the number of domains. To prove the corollary, we just need to prove that $B(1) \geq B(M) \text{ for } M \geq 1 $. Consequently, under the condition, we have:
>     \begin{align}
>         &1 \leq M \leq \frac{C_1\log\delta^{-1}+C_2}{C_3\log2\delta^{-1}} \cdot n \\\\
>         \Rightarrow &\frac{C_3\log{2\delta^{-1}}}{n}(1-M) - \frac{C_1\log\delta^{-1}+C_2}{M}(1-M) \geq 0 \\\\
>         \Rightarrow&B(1) \geq B(M) \quad (M \geq 1)\nonumber
>     \end{align}
> Based on Collary 1, we **design a multi-domain circuits datasets to train the GNN model for enhanced generalization capability**. In this paper, circuits with **similar functionalities**, such as arithmetic, control, and memory, are grouped into distinct circuit domains.
>
> **Moreover, the GNN model achieves high generalization capability by learning domain-invariant information (i.e., inductive bias).** Specifically, [1] observed that the effectiveness of a node-level transformation is closely linked to the local subgraph rooted at the node, regardless of the circuit to which the node belongs. Based on this observation, they proposed extracting subgraphs rooted at individual nodes to generate node embeddings that capture inductive biases. These embeddings are capable of learning domain-invariant representations, thereby enabling the model to generalize effectively to unseen circuits. In this work, we follow [1] to train a GNN model with a strong inductive bias.

---

> ### Author Response · Authors · 2024-11-22
>
> **Step 2. We empirically show that our GESD framework can enhance different kinds of symbolic learning methods.**
> To evaluate **whether GESD can effectively enhance symbolic learning methods**, we combine GESD with three different symbolic learning methods. Specifically, we select classical genetic programming-based SR method --- **GPLearn [2]**, a state-of-the-art (SOTA) deep learning-based SR method --- **DSR [3]**, and an MCTS-based SR method --- **CMO (ours)** as backend symbolic learning methods. The results in Table a show that **our GESD significantly improves the generalization performance of all symbolic learning methods on six challenging open-source circuits.**
>
> **Table a:** G-X means combining GNN and the X symbolic learning method. The results demonstrate the effectiveness of our GESD framework can effectively enhance different kinds of symbolic learning methods.
>  Open-source Circuits | Hyp    | Multiplier | Square | Desperf | Ethernet | Conmax | Average
> :--------------------:|:------:|:----------:|:------:|:-------:|:--------:|:------:|:--------:
>  Method               | Recall | Recall     | Recall | Recall  | Recall   | Recall | Recall
>  G-DSR                | **0.90**   | **0.92**       | **0.87**   | **0.85**   | **0.98**     | **0.84**   | **0.89**
>  DSR                  | 0.20   | 0.11       | 0.46   | 0.76    | 0.72     |0.88   | 0.52
>  G-GPLearn            | **0.64**   | **0.92**       | **0.91**   | **0.80**    | **0.02**     | **0.72**   | **0.67**
>  GPLearn              | 0.35   | 0.11       | 0.27   | 0.39    | 0.02     | 1.00   | 0.36
>  CMO                  | **0.99**   | **0.97**       | **0.98**   | **0.80**    | **0.72**     | **0.85**   | **0.89**
>  CMO without GESD     | 0.93   | 0.52       | 0.72   | 0.60    | 0.42     | 0.45   | 0.61
>
>
> **Step 3. We analyze how the symbolic learning method benefits from the GESD framework.**
> Our GESD **enhances the generalization capability** of the symbolic function **by transferring the domain-invariant information (i.e., inductive bias) learned by the graph model into the symbolic function.** Specifically, we have shown in Step 1 that the teacher GNN achieves high generalization capability through capturing domain-invariant information from well-constructed circuit subgraphs. To verify whether our GESD effectively incorporates this information into the symbolic searching process, we compute the **KL divergence** between the soft labels generated by the teacher GNN and the outputs of the learned symbolic functions. The results in **Table b** demonstrate that **our GESD enables the student symbolic learning method to effectively learn and compensate for the missing inductive bias.** Therefore, the GESD helps address the circuit symbolic generalization problem.
>
> **Table b:** We compute the KL divergence between the teacher GNN model and two variants: our CMO and CMO without GESD. The results reveal that the KL divergence between the GNN and CMO is significantly smaller than that between the GNN and CMO without GESD, demonstrating GESD's effectiveness in distilling inductive bias.
> |                     | Hyp           | Multiplier    | Square         |
> |----------------------|---------------|---------------|----------------|
> | Method               | KL divergence | KL divergence | KL divergence  |
> | CMO              | **0.145**         | **0.473**         | **0.090**          |
> | CMO without GESD | 1.541         | 1.190         | 0.899          |
> |                 |**Desperf**       |**Ethernet**      | **Conmax**      |
> | Method               | KL divergence | KL divergence | KL divergence  |
> | CMO            |**0.08**         | **0.11**         | **0.151**          |
> | CMO without GESD | 0.515         | 0.488         | 0.350          |

---

> > ### Author Response · Authors · 2024-11-22
> >
> > **Step 4. We explain why we chose MCTS rather than other symbolic learning approaches, such as genetic algorithms and deep learning, as the backend function searching method.**
> > In our CS task, we adopt an MCTS-based symbolic learning method **due to its superior training efficiency and robust search capabilities compared to other kinds of symbolic learning approaches.** Specifically, we compare our CMO with G-GPLearn and G-DSR in terms of the offline prediction recall and training time. The results in Table c demonstrate that **our CMO outperforms G-GPLearn in finding generalizable symbolic function**. Moreover, **although G-DSR achieves a comparable prediction recall to CMO, its training process is notably time-consuming.** Therefore, we select MCTS as the backend function searching method.
> >
> > **Table c:** The results demonstrate that our CMO achieves higher prediction recall than the genetic-based method (G-GPLearn) and shorter training time than a SOTA deep learning-based method (G-DSR) on average.
> > |           | Hyp      |               | Multiplier |               | Square |               |        |                |
> > |-----------|----------|---------------|------------|---------------|--------|---------------|--------|----------------|
> > |           | Recall   | Training Time | Recall     | Training Time | Recall | Training Time |        |                |
> > | G-DSR     | 0.90     | 5926.10       | 0.92       | 10061.73      | 0.87   | 10085.49      |        |                |
> > | G-GPLearn | 0.64     | 915.43        | 0.92       | 1542.91       | 0.91   | 1526.16       |        |                |
> > | CMO       | **0.99**     | 2911.38       | **0.97**       | 5427.95       | **0.98**   | 5525.24       |        |                |
> > |           | **Des_perf** |               | **Ethernet**   |               | **Conmax** |               |        | **Average**        |
> > |           | Recall   | Training Time | Recall     | Training Time | Recall | Training Time | Recall | Training Time  |
> > | G-DSR     | **0.85**     | 9593.55       | **0.98**       | 11305.60      | 0.84   | 16926.33      | 0.89   | 10649.80       |
> > | G-GPLearn | 0.80     | 757.10        | 0.02       | 1452.16       | 0.72   | 1377.46       | 0.67   | 1261.87        |
> > | CMO       | 0.80     | 1885.89       | 0.72       | 2249.19       | **0.85**   | 4293.19       | **0.89**   | **3715.47**        |
> >
> > [1]. Zhihai Wang, et al. A circuit domain generalization framework for efficient logic synthesis in chip design. International Conference on Machine Learning, 2024.
> >
> > [2]. Pedro G Espejo, et al. A survey on the application of genetic programming to classification. IEEE Transactions on Systems, Man, and Cybernetics, Part C (Applications and Reviews), 40(2):121–144, 2009.
> >
> > [3]. Brenden K Petersen, et al. Deep symbolic regression: Recovering mathematical expressions from data via risk-seeking policy gradients. In International Conference on Learning Representations, 2020.
> >
> > ### Weakness 2.
> > > **Some of the writings can be improved, e.g. “However, this approach cannot capture effective information from specific circuit distribution for higher generalization performance” – Is it due to the human-designed nature and lack of adoption of machine learning from existing data?**
> >
> > Thanks for your valuable comments. Yes, the generalization capability of human-designed approaches is **constrained by their inherent nature and the absence of machine learning techniques to leverage existing data.** We have clarified and improved this statement in **Line 61 of the first revision**. For your convenience, we provide the updated text below:
> >
> > In contrast, [4] proposes a human-designed hard-coded mathematical expression as the scoring function, which aligns with human intuition and is thus regarded to be reliable. **However, designing and developing these functions is extremely challenging as it requires extensive expert knowledge. Moreover, this function cannot achieve high generalization performance due to the lack of adoption of machine learning from existing data**, which could significantly degrade the QoR of the optimized circuits.
> >
> > [4]. Xing Li, et al. Effisyn: Efficient logic synthesis with dynamic scoring and pruning. In IEEE/ACM International Conference on Computer-Aided Design (ICCAD). IEEE, 2023.
> >
> > ### Weakness 3
> > > **Some technical errors, e.g. “Specifically, we use mean absolute error and focal loss” yet the equation (4) is an MSE loss.**
> >
> > Thanks for your valuable comments. We have corrected the term "mean absolute error" to "mean squared error" in **Line 1090 of the first revision** to align with Equation (4). We appreciate your attention to detail, which helps improve the accuracy and clarity of our work.

---

> > > ### Author Response · Authors · 2024-11-22
> > >
> > > ### Weakness 4
> > > > **Provide more circuit dataset descriptions, e.g., graph sizes and graph visualizations.**
> > >
> > > We provide **detailed statistics of circuits from two open-source benchmarks and one industrial benchmark in Tables 11, 12, 13, and 14 of the initial submission**. Moreover, we supplement a new subsection, **Appendix D.4 (Line 910 and Figure 7)--- "Visualization of the Circuit Graph"---in the first revision for graph visualizations**.
> > >
> > > Specifically, these circuit statistics include information about PIs (Primary Inputs), POs (Primary Outputs), Latches, Nodes (the number of graph nodes), Edges (the number of graph edges), Cubes, and Lev (Level). **In this paper, we use Nodes, Edges, and Lev to represent the size of the graph.** For your convenience, we provide the meanings of the graph information and statistics in Tables 11, 12, 13, and 14 below.
> > > - The fanins of a node are the nodes providing input to it, whereas the fanouts are the nodes it drives.
> > > - Primary Inputs (PIs) are nodes with no fanins, and Primary Outputs (POs) are a subset of the network’s nodes.
> > > - Latches are specialized nodes used in sequential circuits.
> > > - **Nodes correspond to logic gates in the boolean network, while Edges represent the wires connecting them.**
> > > - Cubes represent subsets of input variables in Boolean functions.
> > > - **Lev refers to the depth of the directed acyclic graph (DAG)**, measured as the maximum number of edges between the PIs and POs.
> > >
> > >
> > >
> > > **Table 11**： A detailed description of circuits from the EPFL benchmark. Nodes denote the sizes of circuits, and Lev denotes the depths of circuits.
> > > | Circuit                | PI   | PO   | Latch | Nodes  | Edge   | Cube   | Lev   |
> > > |------------------------|-------|-------|-------|--------|--------|--------|-------|
> > > | Adder                 | 256   | 129   | 0     | 1020   | 2040   | 1020   | 255   |
> > > | Barrel shifter        | 135   | 128   | 0     | 3336   | 6672   | 3336   | 12    |
> > > | Divisor               | 128   | 128   | 0     | 57247  | 114494 | 57247  | 4372  |
> > > | Hypotenuse            | 256   | 128   | 0     | 214335 | 428670 | 214335 | 24801 |
> > > | Log2                  | 32    | 32    | 0     | 32060  | 64120  | 323060 | 444   |
> > > | Max                   | 512   | 130   | 0     | 2865   | 5730   | 2865   | 287   |
> > > | Multiplier            | 128   | 128   | 0     | 27062  | 54124  | 27062  | 274   |
> > > | Sin                   | 24    | 25    | 0     | 5416   | 10832  | 5416   | 225   |
> > > | Square-root           | 128   | 64    | 0     | 24618  | 49236  | 24618  | 5058  |
> > > | Square                | 64    | 128   | 0     | 18486  | 36969  | 18485  | 250   |
> > > | Round-robin arbiter   | 256   | 129   | 0     | 11839  | 23678  | 11839  | 87    |
> > > | Alu control unit      | 7     | 26    | 0     | 175    | 348    | 174    | 10    |
> > > | Coding-cavlc          | 10    | 11    | 0     | 693    | 1386   | 693    | 16    |
> > > | Decoder               | 8     | 256   | 0     | 304    | 608    | 304    | 3     |
> > > | i2c controller        | 147   | 142   | 0     | 1357   | 2698   | 1356   | 20    |
> > > | Int to float converter| 11    | 7     | 0     | 260    | 520    | 260    | 16    |
> > > | Memory controller     | 1204  | 1230  | 0     | 47110  | 93945  | 47109  | 114   |
> > > | Priority encoder      | 128   | 8     | 0     | 978    | 1956   | 978    | 250   |
> > > | Lookahead XY router   | 60    | 30    | 0     | 284    | 514    | 257    | 54    |
> > > | Voter                 | 1001  | 1     | 0     | 13758  | 27516  | 13758  | 70    |

---

> > > > ### Author Response · Authors · 2024-11-22
> > > >
> > > > **Table 12**: A detailed description of circuits from the IWLS benchmark.
> > > > | Circuit                     | PI   | PO   | Latch  | Nodes   | Edge    | Cube    | Lev   |
> > > > |-----------------------------|-------|-------|--------|---------|---------|---------|-------|
> > > > | aes_core                   | 259   | 129   | 530    | 20797   | 40645   | 24444   | 28    |
> > > > | des_area                   | 240   | 64    | 128    | 5005    | 9882    | 5889    | 35    |
> > > > | des_perf                   | 234   | 64    | 8808   | 98463   | 180542  | 108666  | 28    |
> > > > | ethernet                   | 98    | 115   | 10544  | 46804   | 113378  | 72850   | 37    |
> > > > | i2c                        | 19    | 14    | 128    | 1147    | 2299    | 1375    | 15    |
> > > > | mem_ctrl                   | 115   | 152   | 1083   | 11508   | 26436   | 14603   | 31    |
> > > > | pci_bridge32               | 162   | 207   | 3359   | 16897   | 34607   | 23130   | 29    |
> > > > | pci_conf_cyc_addr_dec      | 32    | 32    | 0      | 109     | 212     | 128     | 6     |
> > > > | pci_spoci_ctrl             | 25    | 13    | 60     | 1271    | 2637    | 1557    | 19    |
> > > > | sasc                       | 16    | 12    | 117    | 552     | 1148    | 766     | 10    |
> > > > | simple_spi                 | 16    | 12    | 132    | 823     | 1694    | 1089    | 14    |
> > > > | spi                        | 47    | 45    | 229    | 3230    | 6904    | 4054    | 32    |
> > > > | steppermotordrive          | 4     | 4     | 25     | 228     | 397     | 253     | 11    |
> > > > | systemcaes                 | 260   | 129   | 670    | 7961    | 18236   | 11648   | 44    |
> > > > | systemcdes                 | 132   | 65    | 190    | 3324    | 6304    | 3791    | 33    |
> > > > | tv80                       | 14    | 32    | 359    | 7166    | 16280   | 9352    | 50    |
> > > > | usb_funct                  | 128   | 121   | 1746   | 12871   | 27102   | 16378   | 25    |
> > > > | usb_phy                    | 15    | 18    | 98     | 559     | 1001    | 638     | 12    |
> > > > | vga_lcd                    | 89    | 109   | 17079  | 124050  | 242332  | 146201  | 25    |
> > > > | wb_conmax                  | 1130  | 1416  | 770    | 29036   | 77185   | 39619   | 26    |
> > > > | wb_dma                     | 217   | 215   | 263    | 3495    | 7052    | 4496    | 26    |
> > > >
> > > > **Table 13**: A detailed description of two very large-scale circuits from the EPFL benchmark
> > > > | Circuit | PI  | PO  | Latch | Nodes     | Lev |
> > > > |---------|------|------|-------|-----------|-----|
> > > > | twenty  | 137  | 60   | 0     | 20732893  | 162 |
> > > > | sixteen | 117  | 50   | 0     | 16216836  | 140 |
> > > >
> > > > **Table 14**: A statistical description of 27 industrial circuits (23 training circuits and 4 testing circuits) from Huawei HiSilicon.
> > > > | Circuit Type       | Metric | PI       | PO       | Latch | Nodes    | Lev       |
> > > > |--------------------|--------|----------|----------|-------|----------|-----------|
> > > > | **Training Circuits** | mean   | 8410.5   | 5978.68 | 0     | 104229.4 | 55.95  |
> > > > |                    | max    | 59974    | 29721    | 0     | 788288   | 104       |
> > > > |                    | min    | 41       | 107      | 0     | 2775     | 18        |
> > > > | **Testing Circuits**  | mean   | 18540.67 | 18015    | 0     | 356111.2 | 103.33  |
> > > > |                    | max    | 42257    | 33849    | 0     | 655243   | 185       |
> > > > |                    | min    | 523      | 483      | 0     | 24778    | 40        |
> > > >
> > > > To provide a more comprehensive understanding of the circuit graph, we have included a new subsection, **Appendix D.4 (Line 912 and Figure 7)---"Visualization of the Circuit Graph"---in the first revision**. In this subsection, **we visualize the Boolean network of a small circuit across different phases of circuit optimization and demonstrate how the CS heuristics drive the optimization process.** For your convenience, we provide the relevant context below:
> > > >
> > > > **Visualization of The Circuit Graph** In the CS stage, a circuit is usually modeled by a DAG. Common types of DAGs for CS include And-Inverter Graphs (AIGs) for pre-mapping optimization and K-Input Look-Up Tables (K-LUTs) for post-mapping optimization. In the pre-mapping optimization phase, an AIG is a DAG containing four types of nodes: the constant, PIs, POs, and two-input And (And2) nodes. A graph edge is either complemented or not. A complemented edge indicates that the signal is complemented. In the post-mapping optimization phase, a K-LUT is a DAG with nodes corresponding to Look-Up Tables and directed edges corresponding to wires. A Look-Up Table in a K-LUT is a digital memory that implements the Boolean function of the node. **To further illustrate the circuit graph, we visualize the AIG, K-LUT look-up table, and the circuit optimization process of a small circuit selected from IWLS2020 [5] in Figure 7.**
> > > >
> > > > [5]. Shubham Rai, et al. Logic synthesis meets machine learning: Trading exactness for generalization. In 2021 Design, Automation & Test in Europe Conference & Exhibition (DATE), pp. 1026–1031. IEEE, 2021.

---

> > > > > ### Author Response · Authors · 2024-11-24
> > > > > **We would love to hear your feedback**
> > > > >
> > > > > Dear Reviewer oLyS,
> > > > >
> > > > > We greatly appreciate your careful reading and constructive comments! We sincerely hope that our rebuttal **has properly addressed all your concerns**, including **deeper explanation and theoretical analysis on why combining GNN, MCTS, and symbolic learning leads to better results** (see *Weakness 1* in rebuttal), and **More circuit dataset descriptions** (see *Weakness 4* in rebuttal). Item-by-item responses to your comments are provided above this response for your reference.
> > > > >
> > > > > As the deadline for the author-reviewer discussion period is approaching (due on Novemeber 27), and **we are looking forward to your feedback and/or questions**! We would deeply appreciate it if you could raise your score if our rebuttal has addressed your concerns. If not, please let us know your further concerns, and we will continue actively responding to your comments and improving our submission.
> > > > >
> > > > > Best,
> > > > >
> > > > > Authors

---

> > > > > > ### Author Response · Authors · 2024-11-28
> > > > > > **We eagerly await your feedback**
> > > > > >
> > > > > > Dear Reviewer oLyS,
> > > > > >
> > > > > > We are writing to gently remind you that **the deadline for the author-reviewer discussion period is approaching** (due on December 2nd). We eagerly await your feedback to understand if our responses have adequately addressed all your concerns. *If so, we would deeply appreciate it if you could raise your score*. If not, please let us know your further concerns, and we will continue actively responding to your comments and improving our submission. We sincerely thank you once more for your insightful comments and kind support.
> > > > > >
> > > > > > Best,
> > > > > >
> > > > > > Authors

---

> > > > > > > ### Author Response · Authors · 2024-11-30
> > > > > > > **We would greatly appreciate hearing your feedback.**
> > > > > > >
> > > > > > > Dear Reviewer oLyS,
> > > > > > >
> > > > > > > We are writing to kindly remind you that the deadline for the author-reviewer discussion period is fast approaching (**ending in two days** on December 2nd). We greatly value your feedback and are eager to hear your thoughts on whether our responses have sufficiently addressed your concerns. Thank you once again for your thoughtful comments and valuable support throughout this process.
> > > > > > >
> > > > > > > Best,
> > > > > > >
> > > > > > > Authors

---

> > > > > > > > ### Author Response · Authors · 2024-12-02
> > > > > > > > **We would greatly appreciate hearing your feedback.**
> > > > > > > >
> > > > > > > > Dear Reviewer oLyS,
> > > > > > > >
> > > > > > > > We would like to express our sincere gratitude once again for your positive feedback, insightful comments, and constructive suggestions. Your guidance has been invaluable in helping us improve the quality of our work!
> > > > > > > >
> > > > > > > > We are writing to gently remind you that **the author-reviewer discussion period will end in less than 36 hours**. We eagerly await your feedback to **understand if our responses have adequately addressed your concerns**. **If so, we would deeply appreciate it if you could raise your score**. If not, we are eager to address any additional queries you might have, which will enable us to further enhance our work.
> > > > > > > >
> > > > > > > > Once again, thank you for your kind support and constructive suggestions!
> > > > > > > >
> > > > > > > > Best,
> > > > > > > >
> > > > > > > > Authors

---

> > > > > > > > > ### Comment · Reviewer_oLyS · 2024-12-02
> > > > > > > > > **Thank you**
> > > > > > > > >
> > > > > > > > > I appreciate authors' response and would like to raise my overall rating to 7.

---

> ### Author Response · Authors · 2024-12-03
> **Appreciation for Your Feedback**
>
> Dear Reviewer oLys,
>
> We are truly grateful for your kind support and the time you've taken to review and assess our work. Your willingness to consider raising the score means a great deal to us and gives us further confidence in the value of our contributions.
>
> We would like to gently remind you that **ICLR's scoring system does not include a score of 7, and the next available score above 6 is 8**. In other AI conferences such as NeurIPS and ICML, **a score of 7** typically corresponds to an **"accept"**, which **is equivalent to a score of 8 at ICLR**. Therefore, **if our responses have adequately addressed your concerns, we sincerely hope you might consider raising the score to 8 (accept)** within the available range. We summarize our responses below.
>
> - **Theoretical and empirical analysis of combining GNN, MCTS, and symbolic learning.** We provide both theoretical and empirical evidence to justify the integration of GNN, MCTS, and symbolic learning in our method. **Theoretically**, we demonstrate that **leveraging multi-domain circuit training datasets** significantly **reduces the generalization error bound** of the teacher GNN model. **Empirically**, we show that the Graph Enhanced Symbolic Discovery (GESD) framework, which combines the teacher GNN with a student symbolic learning model, enhances the generalization ability of the symbolic function by **transferring domain-invariant knowledge (i.e., inductive bias)** from the graph model to the symbolic function. **Additionally**, our experiments highlight the **superior performance of MCTS** over classical symbolic learning methods **in terms of both generalization capability and training efficiency**. These analyses illustrate the rationale for integrating GNN, MCTS, and symbolic learning. Moreover, **this combination enables our method to discover a generalizable, lightweight, and interpretable symbolic function.**
>
> - **Some of the writings can be improved.** We sincerely appreciate your valuable feedback on improving the clarity of our writing. **We have carefully revised the unclear statements in the revised version.**
>
> - **Technical errors.** We sincerely appreciate you for pointing out the technical errors and **we have carefully addressed and corrected in the revised version**.
>
> - **More circuit dataset descriptions and graph visualization.** We have included **detailed statistics for circuits** from two open-source and one industrial benchmark in Tables 11–14 of the rebuttal. Additionally, we added **graph visualizations in the revised version** under Appendix D.4 (Line 910, Figure 7) titled "Visualization of the Circuit Graph."
>
> Once again, thank you for your constructive suggestions and for considering our work so thoughtfully. Your feedback has been instrumental in helping us improve!
>
> Best,
>
> Authors

---

> > ### Author Response · Authors · 2024-12-03
> > **We are looking forward to your further feedback**
> >
> > Dear Reviewer oLys,
> >
> > We sincerely appreciate your thoughtful engagement and your kind consideration of raising the score for our work—it truly encourages us and reinforces our confidence in the value of our contributions.
> >
> > As mentioned earlier, ICLR’s scoring system **does not include a score of 7, and the next available score above 6 is 8**, which corresponds to an **"accept"** decision. With the discussion period nearing its conclusion in **less than 5 hours**, we wanted to kindly check if our responses have addressed your concerns effectively. **If so, we would be most grateful if you might consider reflecting this in your final assessment**. If you have any additional questions or suggestions, please don’t hesitate to let us know—we would be delighted to provide further clarifications promptly within the remaining time.
> >
> > Thank you again for your valuable feedback and support throughout this process!
> >
> > Best,
> >
> > Authors

---

> > > ### Author Response · Authors · 2024-12-04
> > > **Thank you!**
> > >
> > > Thank you for your kind support and valuable feedback on our paper! We appreciate your insightful comments and constructive suggestions.

---

### Official Review · Reviewer_s1pC · 2024-11-04

**Soundness:** 4
**Presentation:** 3
**Contribution:** 3
**Rating:** 6
**Confidence:** 3

**Summary:**

This authors propose a novel data-driven circuit symbolic learning framework, CMO. It learns a symbolic scoring function balancing inference efficiency, interpretability, and generalization performance. While existing approaches often struggle with these trade-offs in modern circuit synthesis (CS) tools, CMO demonstrates superior capability in discovering lightweight and interpretable symbolic functions from a decomposed symbolic space. The major technical contribution of CMO is the Graph Enhanced Symbolic Discovery (GESD) framework, which employs a specially designed Graph Neural Network (GNN) to guide the generation of symbolic trees. CMO is the first graph-enhanced approach for discovering lightweight and interpretable symbolic functions that effectively generalize to unseen circuits in CS.

**Strengths:**

1. Overall, the proposed work is well-structured with a profound related work.
2. This paper proposes a novel circuit symbolic learning framework to learn efficient, interpretable, and generalizable symbolic functions that are reliable and simple to deploy in modern CS tools.
3. CMO is the first graph-enhanced approach for discovering lightweight and interpretable symbolic functions that can well generalize to unseen circuits in CS.
4. Extensive experimental results show the effectiveness of the proposed CMO over existing works.

**Weaknesses:**

The link between the two methods in sections 4.1 and 4.2 needs to be further elucidated, and it is not currently possible to visualize in the text the specific interrelationships between the two methods. For example, what is the role of si in section 4.1 in section 4.2 and what is the flow of the calculations for si.

**Questions:**

Please check weakness

---

> ### Author Response · Authors · 2024-11-22
> **Rebuttal by Authors**
>
> # Response to Reviewer s1pC
> We thank the reviewer for the insightful and valuable comments. We respond to each comment as follows and sincerely hope that our rebuttal can properly address your concerns. If so, we would deeply appreciate it if you could raise your score. If not, please let us know your further concerns, and we will continue actively responding to your comments and improving our submission.
>
> ### Weakness 1.
> > **The link between the two methods in sections 4.1 and 4.2 needs to be further elucidated, and it is not currently possible to visualize in the text the specific interrelationships between the two methods.**
>
> Thanks for your valuable comments. The relationship between the two approaches described in Sections 4.1 and 4.2 is as follows: **The GESD framework discussed in Section 4.2 is the detailed description of the structural and semantic function learning component outlined in Figure 2 of Section 4.1 .** To provide clearer clarification of the connection between Sections 4.1 and 4.2, we have updated Figure 2 by changing the label 'semantic/structural function learning' to **'GESD for semantic/structural function learning.'** Additionally, we **added a textual description in Section 4.1 of the first revision (line 241) to explain how GESD works in our CMO framework.** For your convenience, we have included the relevant supplementary content from Section 4.1 below.
>
> **GESD for Symbolic Function Learning**
> After decomposing the init feature into structural and semantic components, we collect structural data $\mathcal{D}\_{str} = \lbrace\textbf{x}\_i^{str}, y_i\rbrace_{i=1}^N $ and semantic data $\mathcal{D}\_{sem} = \lbrace\textbf{x}\_i^{sem}, y_i\rbrace_{i=1}^N $, where $\textbf{x}\_i^{str} $ refers to structural node feature and $\textbf{x}\_i^{sem} $ refers to semantic node feature. To capture structural information, we employ our Graph Enhanced Symbolic Discovery (GESD) framework to learn a mathematical symbolic function $f^{str}: \mathbf{R}^{d} \to \mathbf{R}  $ (see Section 4.2), as the values of structural features can be approximated as continuous data, making them suitable for continuous mathematical symbolic regression.  In contrast, learning mathematical functions for semantic information is challenging due to the discrete and binary nature of both feature values and labels. Thus, to capture semantic information, we formulate the semantic function as a Boolean symbolic learning problem, i.e., employing our GESD framework to learn a boolean function $f^{sem}: \mathbf{B}^{d} \to \mathbf{B} $ (see Section 4.2) that can accurately identify the effective nodes, where $\mathbf{B} = \lbrace 0,1 \rbrace $ denotes the boolean feature domain.

---

> > ### Author Response · Authors · 2024-11-22
> >
> > ### Weakness 2.
> > > **What is the role of si in sections 4.1 and 4.2 and what is the flow of the calculations for si?**
> >
> > $\textbf{s} = \lbrace s_i \rbrace \_{i=1}^N $ are **calculated to predict ineffective nodes on a circuit with $N $ nodes and avoid transformation on these nodes to accelerate the CS heuristics in the online phase**. The lower the score, the higher the probability that the node is ineffective.
> >
> > The calculations of $\textbf{s} $ consist of two steps: **(a). collect training dataset $\mathcal{D} = \lbrace \textbf{x}\_i, y_i \rbrace_{i=1}^N $ and train a pair of structural function $f_{str} $ and semantic function $f_{sem} $ in the offline phase; (b). leveraging the symbolic functions to calculate the score $s_i = f_{str}(x_i^{str}) + w * f_{sem}(x_i^{sem}) $ for all nodes $x_i^{test} = [x_i^{str}, x_i^{sem}] $ on an unseen circuit in the online phase.** The detailed offline training and online score calculation algorithm are provided **in Algorithms 1, 2, and 3 of our first revision**. For your convenience, we also provide the Python pseudocode for them below:
> >
> > ```python
> > # The offline training algorithm
> > def training(D, f_GNN):
> >     """
> >     Input:
> >         Training dataset D
> >         The trained GNN model f_GNN
> >     Output:
> >         Semantic function f_sem, Structural function f_str
> >     """
> >
> >     # Step 1: Separate the initial dataset into structural and semantic data
> >     D_str = extract_structural_data(D)  # Extract structural data from D
> >     D_sem = extract_semantic_data(D)    # Extract semantic data from D
> >
> >     # Step 2: Learn the structural function using GESD
> >     f_str = GESD(D_str, f_GNN)
> >
> >     # Step 3: Learn the semantic function using GESD
> >     f_sem = GESD(D_sem, f_GNN)
> >
> >     return f_str, f_sem
> > ```
> > ```python
> > # The GESD algorithm
> > def GESD(D, f_GNN)
> >     """
> >
> >     Input:
> >         The structural/semantic training dataset D
> >         The trained GNN model f_GNN
> >     Output:
> >         Optimal function f*
> >     """
> >
> >     for episode in range(num_episodes):  # Repeat for each episode
> >         # Selection phase
> >         s_t = []  # Initialize state as empty
> >         t = 0  # Initialize step counter
> >         while is_expandable(s_t) and t < t_max:
> >             a_t_plus_1 = select_action_with_uct(s_t)  # Select action using UCT
> >             s_next, NT = take_action(s_t, a_t_plus_1)  # Perform action and observe next state
> >             s_t = s_next  # Update state
> >             mark_as_visited(s_t)  # Mark state as visited
> >             t += 1  # Increment step counter
> >
> >         # Expansion phase
> >         a = select_unvisited_action(s_t)  # Randomly select an unvisited action
> >         s_next = take_action_randomly(s_t, a)  # Perform action and observe next state
> >         s_t = s_next  # Update state
> >         mark_as_visited(s_t)  # Mark state as visited
> >         t += 1  # Increment step counter
> >
> >         # Simulation phase
> >         for _ in range(num_simulations):  # Run multiple simulations
> >             s_sim = s_t  # Fix the starting point
> >             t_sim = t  # Initialize simulation step counter
> >             while not is_terminal(s_sim) and t_sim < t_max:
> >                 a_sim = select_random_action(s_sim)  # Randomly select an action
> >                 s_next_sim = take_action_randomly(s_sim, a_sim)  # Perform action
> >                 s_sim = s_next_sim  # Update state
> >                 t_sim += 1  # Increment simulation step counter
> >
> >             if forms_complete_expression_tree(s_sim):  # Check if expression tree is complete
> >                 f = project_function(s_sim)  # Project function
> >                 y_pred = f(D.x)  # Calculate function prediction
> >                 z_pred = f_GNN(D.x)  # Calculate GNN prediction
> >                 reward = calculate_reward(
> >                     y_pred, D.y, z_pred, lambda_param, eta
> >                 )  # Compute reward
> >
> >         # Backpropagation phase
> >         backpropagate_simulation_results(s_t, reward)  # Propagate simulation results and update counts
> >
> >     return f*  # Return the optimal function
> > ```
> > ```python
> > # The online scores calculation algorithm
> > def inference_process(D_test, f_str, f_sem):
> >     """
> >     Input:
> >         test_dataset D_test
> >         structural_function f_str
> >         semantic_function f_sem
> >     Output:
> >         scores s: final scores for all nodes in the test dataset
> >     """
> >
> >     # Step 1: Separate the initial dataset into structural and semantic data
> >     D_str = extract_structural_data(D_test)  # Extract structural data from D
> >     D_sem = extract_semantic_data(D_test)    # Extract semantic data from D
> >
> >     # Step 2: Calculate structural and semantic scores
> >     s_str = [
> >         f_str(x_str) for x_str, _ in D_str
> >     ]
> >     semantic_scores = [
> >         f_sem(x_sem) for x_sem, _ in D_sem
> >     ]
> >
> >     # Step 3: Calculate the weight as the median of structural scores
> >     w = median(s_str)
> >
> >     # Step 4: Calculate final scores
> >     s = [
> >         s_str^i + weight * s_sem^i
> >         for s_str, s_sem in zip(s_str, s_sem)
> >     ]
> >
> >     return final_scores
> > ```

---

> > > ### Author Response · Authors · 2024-11-24
> > > **We would love to hear your feedback**
> > >
> > > Dear Reviewer s1pC,
> > >
> > > We greatly appreciate your careful reading and constructive comments! We sincerely hope that our rebuttal **has properly addressed all your concerns**, including **visualizing in the text the specific interrelationships between the two method** (see *Weakness 1* in rebuttal), and **illustration of the calculations flow of $s_i $** (see *Weakness 2* in rebuttal). Item-by-item responses to your comments are provided above this response for your reference.
> > >
> > > As the deadline for the author-reviewer discussion period is approaching (due on Novemeber 27), and **we are looking forward to your feedback and/or questions**! We would deeply appreciate it if you could raise your score if our rebuttal has addressed your concerns. If not, please let us know your further concerns, and we will continue actively responding to your comments and improving our submission.
> > >
> > > Best,
> > >
> > > Authors

---

> > > > ### Author Response · Authors · 2024-11-28
> > > > **We eagerly await your feedback**
> > > >
> > > > Dear Reviewer s1pC,
> > > >
> > > > We are writing to gently remind you that **the deadline for the author-reviewer discussion period is approaching** (due on December 2nd). We eagerly await your feedback to understand if our responses have adequately addressed all your concerns. *If so, we would deeply appreciate it if you could raise your score*. If not, please let us know your further concerns, and we will continue actively responding to your comments and improving our submission. We sincerely thank you once more for your insightful comments and kind support.
> > > >
> > > > Best,
> > > >
> > > > Authors

---

> > > > > ### Author Response · Authors · 2024-11-30
> > > > > **We would greatly appreciate hearing your feedback.**
> > > > >
> > > > > Dear Reviewer s1pC,
> > > > >
> > > > > We are writing to kindly remind you that the deadline for the author-reviewer discussion period is fast approaching (**ending in two days** on December 2nd). We greatly value your feedback and are eager to hear your thoughts on whether our responses have sufficiently addressed your concerns. Thank you once again for your thoughtful comments and valuable support throughout this process.
> > > > >
> > > > > Best
> > > > >
> > > > > Authors

---

> > > > > > ### Author Response · Authors · 2024-12-02
> > > > > > **We would greatly appreciate hearing your feedback.**
> > > > > >
> > > > > > Dear Reviewer s1pC,
> > > > > >
> > > > > > We would like to express our sincere gratitude once again for your positive feedback, insightful comments, and constructive suggestions. Your guidance has been invaluable in helping us improve the quality of our work!
> > > > > >
> > > > > > We are writing to gently remind you that **the author-reviewer discussion period will end in less than 36 hours**. We eagerly await your feedback to **understand if our responses have adequately addressed your concerns**. **If so, we would deeply appreciate it if you could raise your score**. If not, we are eager to address any additional queries you might have, which will enable us to further enhance our work.
> > > > > >
> > > > > > Once again, thank you for your kind support and constructive suggestions!
> > > > > >
> > > > > > Best,
> > > > > >
> > > > > > Authors

---

> > > > > > > ### Author Response · Authors · 2024-12-03
> > > > > > > **We would greatly appreciate hearing your feedback**
> > > > > > >
> > > > > > > Dear Reviewer s1pC,
> > > > > > >
> > > > > > > We would like to sincerely thank you once again for your positive feedback, insightful comments, and constructive suggestions. Your guidance has been invaluable in enhancing the quality of our work!
> > > > > > >
> > > > > > > As the author-reviewer discussion period is approaching its conclusion with **less than 12 hours** remaining, we wanted to kindly follow up regarding your feedback. We are eager to hear your thoughts on **whether our responses have sufficiently addressed your concerns**. **If they have, we would greatly appreciate it if you could consider reflecting this in your score.** If there are any remaining questions or concerns, we would be more than happy to provide further clarifications within the remaining time.
> > > > > > >
> > > > > > > Thank you again for your support and thoughtful guidance throughout this process. We deeply value your time and effort in reviewing our work.
> > > > > > >
> > > > > > > Best，
> > > > > > >
> > > > > > > Authors

---

> > > > > > > > ### Author Response · Authors · 2024-12-03
> > > > > > > > **We would greatly appreciate hearing your feedback**
> > > > > > > >
> > > > > > > > Dear Reviewer s1pC,
> > > > > > > >
> > > > > > > > We would like to sincerely thank you once again for your positive feedback, insightful comments, and constructive suggestions. Your guidance has been instrumental in improving the quality of our work.
> > > > > > > >
> > > > > > > > As the author-reviewer discussion period enters its final hours with **less than 5 hours** remaining, we wanted to kindly follow up regarding your feedback. We would greatly value your thoughts on **whether our responses have sufficiently addressed your concerns**. **If so, we would deeply appreciate it if you could consider reflecting this in your score**. If there are any remaining questions or concerns, we would be more than happy to provide further clarifications within the remaining time.

---

> > > > > > > > > ### Author Response · Authors · 2024-12-03
> > > > > > > > > **We would greatly appreciate hearing your feedback**
> > > > > > > > >
> > > > > > > > > Dear Reviewer s1pC,
> > > > > > > > >
> > > > > > > > > We sincerely thank you once again for your positive feedback, insightful comments, and constructive suggestions. Your guidance has been invaluable in enhancing the quality of our work.
> > > > > > > > >
> > > > > > > > > As the author-reviewer discussion period enters its final stage, with **less than two hours** remaining, we wanted to kindly follow up on your feedback. We would greatly appreciate your thoughts on **whether our responses have adequately addressed your concerns**. **If they have, we would be truly grateful if you could consider reflecting this in your score**. Should you have any remaining questions or concerns, we would be more than happy to provide further clarifications within the remaining time.

---

> > > > > > > > > > ### Author Response · Authors · 2024-12-04
> > > > > > > > > > **Thank you!**
> > > > > > > > > >
> > > > > > > > > > Thank you for your kind support and valuable feedback on our paper! We appreciate your insightful comments and constructive suggestions.

---

### Author Response · Authors · 2024-12-04
**Summary of our responses**

Dear Area Chair,

We are writing as the authors of the paper "A Graph Enhanced Symbolic Discovery Framework for Efficient Logic Synthesis" (ID: 7225). We would like to express our sincere gratitude for your dedication and support throughout the review process.

The reviewers rated our work as **8** Accept (Reviewer kVA7), **6 (raised to 7)** Weak Accept (Reviewer oLys), and **6** Weak accept (Reviewer s1pC), respectively.

For your convenience, we **have prepared a summary of our responses** and outlined how we have addressed the reviewers' concerns as follows. We sincerely hope that this summary will facilitate your review and lighten your workload. Thank you once again for your time and support.

### Summary of our responses
Our paper has received encouraging positive feedbacks from the reviewers, such as **"the paper is well-structured"** (Reviewer s1pC), **"a profound related work"** (Reviewer s1pC), **"well-written"** (Reviewer oLyS), **"Choice of benchmarks and heuristics are solid"** (Reviewer oLyS), **"achieves significant speedup"** (Reviewer oLyS), **"clearly introduced"** (Reviewer kVA7) and **"comprehensive experiments"** (Reviewer kVA7 and s1pC).

**Reviewer kVA7** has replied that "My main concerns previously were the misused terms, vague details, and the lack of baselines. **Most of them have been addressed** in the rebuttal or revised in the manuscript by the authors. Thus, I decided to **raise my score to 7/8**. I hope the authors can retain these revisions in the manuscript and make them clear.".

**Reviewer oLys** has replied that "I appreciate authors' response and would like to raise my **overall rating to 7**." While we regret that ICLR does not include a score of 7 in its rating scale, we humbly believe this indicates **the reviewer's propensity to accept the paper**.

**Reviewer s1pC** has **provided many positive comments** on our work, such as "well-structured", "profound related work" and "comprehensive expreiments". While we regret not receiving a response to our follow-up, we humbly believe that we have effectively addressed the reviewer’s primary concerns. Below, we summarize how we have addressed the reviewer's feedback.

- **Clarification for the interrelationships between the two Sections in method.** We have clarified that the two methods are presented **in a progressive relationship**. Section 4.2 provides a detailed explanation of the method introduced in Section 4.1. In the revised version, we’ve added both textual explanations and diagrams to clearly illustrate this progression.

- **Explanation for the role and calculations flow of the score $s_i $.** We have explained that the score $s_i $ is calculated to **predict and prune ineffective nodes** in an unseen circuit, thereby **accelerating the CS heuristics**. In the revised version, we’ve added three detailed algorithms that outline the step-by-step calculation process of $s_i $.

Once again, thank you very much for your time and efforts throughout the review period.

Best,

Authors

---

### Meta-Review · Area_Chair_vXri · 2024-12-17

**Metareview:**

This paper studies the problem of Logic Optimization, where the goal is to optimize circuits. The paper proposes a new data-driven symbolic learning framework (CMO). The proposed  method trains a GNN and uses an MCTS-based symbolic regression method to generate symbolic scoring functions, ensuring both inference efficiency and generalization. Overall the reviewers found the paper to be well-written and the experiments are convincing. There were some concerns (such as misuse of terms) but most of them were addressed in the response period.

**Additional Comments On Reviewer Discussion:**

During the discussion periods the reviewers acknowledged that the new version has fixed some of their concerns.

---

### Decision · Program_Chairs · 2025-01-22

Accept (Poster)